# Linking spontaneous and stimulated spine dynamics

Maximilian F. Eggl [1,6], Thomas E. Chater[2,3,6], Janko Petkovic[1,6], Yukiko Goda[2,4] &
Tatjana Tchumatchenko [1,5 ✉]

Our brains continuously acquire and store memories through synaptic plasticity. However, spontaneous synaptic changes can also occur and pose a challenge for maintaining stable memories. Despite fluctuations in synapse size, recent studies have shown that key population-level synaptic properties remain stable over time. This raises the question of how local synaptic plasticity affects the global population-level synaptic size distribution and whether individual synapses undergoing plasticity escape the stable distribution to encode specific memories. To address this question, we *(i)* studied spontaneously evolving spines and *(ii)* induced synaptic potentiation at selected sites while observing the spine distribution pre- and post-stimulation. We designed a stochastic model to describe how the current size of a synapse affects its future size under baseline and stimulation conditions and how these local effects give rise to population-level synaptic shifts. Our study offers insights into how seemingly spontaneous synaptic fluctuations and local plasticity both contribute to population-level synaptic dynamics.

[1] University of Mainz Medical Center, Anselm-Franz-von-Bentzel-Weg 3, 55128 Mainz, Germany. [2] Laboratory for Synaptic Plasticity and Connectivity, RIKEN Center for Brain Science, Wako-shi, Saitama, Japan. [3] Department of Physiology, Keio University School of Medicine, Tokyo, Japan. [4] Synapse Biology Unit, Okinawa Institute of Science and Technology Graduate University, Onna-son, Kunigami-gun, Okinawa, Japan. [5] Institute of Experimental Epileptology and Cognition Research, University of Bonn Medical Center, Venusberg-Campus 1, 53127 Bonn, Germany. [6] These authors contributed equally: Maximilian F. Eggl, Thomas E. Chater, Janko Petkovic. ✉email: tatjana.tchumatchenko@uni-bonn.de

Memory and learning are thought to rely on changes in synaptic strength, characterized by the strengthening and weakening of specific synaptic connections[1–6]. Several studies have targeted the molecular mechanisms of synaptic plasticity both on short time scales[7] and on the time scales of hours or even days[8–10].

While synaptic plasticity is often directed at specific synaptic sites, synapses can also be dynamic in the absence of directed plasticity, and disentangling spontaneous from directed synaptic changes can be challenging[11]. Synapses undergo significant size changes over hours and days, most likely driven by spontaneous dynamics of synaptic molecules[12–18]. Despite each synapse being subject to potentially large fluctuations over time, average population features show remarkable stability in time[13,15,19–24].

Many experimentally reported synaptic size distributions are asymmetric and exhibit a long right tail, which has been hypothesized to be linked to optimality with respect to information storage capacity, neuronal firing rates, and long-distance information transfer[25,26]. While it is commonly assumed that these distributions arise from the cumulative action of spines shrinking and growing[23,27], the interaction between activity-independent and activity-dependent components is not fully understood[24].

Additionally, modeling studies often make one essential assumption: synapses retain their properties indefinitely when not actively driven to change. This assumption is fundamental because otherwise, spontaneously occurring changes would lead to modifications in the network function or unlearning newly acquired skills. However, the fact that synaptic changes are driven by molecular processes that are inherently noisy (e.g., lateral diffusion, active trafficking, endocytosis, and exocytosis[17,28]) implies that such spontaneous changes are inevitable. Thus, studying how fundamental characteristics of the synapse populations are retained (e.g., probability of release, total receptor conductance, size, morphology, ultrastructure, composition) over longer time scales is another crucial aspect of understanding memory. This capacity of the synapses to retain their features is known in the literature as synaptic tenacity[11,29].

Models linking these findings to single spine dynamics using various approaches already exist[12,15,22,30,31]. In this study, we introduce a model that can reproduce both long-term potentiation (LTP)-triggered spine changes and activity-independent spine fluctuations within a common framework. It is hypothesized that LTP impacts small spines more because they have more room to grow[32], while larger spines could represent stable long-term memory storage[33,34]. Within the activity-independent context, it has been shown that large spines vary more[12,15,24].

Our model, which is inspired by the Kesten process and the multiplicative dynamics of previous studies, allowed us to recreate the experimental results relating to spontaneous spine fluctuations while relying their log-normal nature. We also were able to use our model to describe spines after LTP induction and report a distinct increase in entropy (a measure of the capacity of a dendrite to store information). Our results describing the spontaneous spine fluctuations are consistent with previously reported effects such as the variance of the large spines, stable population distribution, and the oscillatory behavior of the spines due to a negative correlation between timesteps[12,13,31] and can explain how LTP signals impact the spontaneous spine distributions.

## Results

We hypothesize that a baseline process that gives rise to the spontaneous spine distribution (activity-independent spine plasticity) is modified by plasticity induction such that both spontaneous and induced spine distributions can be described using the same model with different model states. Therefore, before considering the stimulation effects, we wanted to understand the model mechanisms needed to capture the activity-independent, spontaneous spine fluctuations.

To this end, we imaged spines on apical oblique dendrites of GFP-expressing CA1 pyramidal neurons in cultured hippocampal organotypic slices. For one set of experiments, we quasi-simultaneously potentiated a subset of spines using glutamate uncaging to induce structural LTP (sLTP) (the activity-dependent or stimulation set, see "Methods" and Supplementary Fig. 1). In another independent set of experiments, the caged glutamate molecule was omitted from the bath, and thus spines did not undergo sLTP following laser illumination. This sham stimulation dataset acted as our activity-independent set. For both cases, over 55 min (15 min pre-stimulation and 40 min post-stimulation), we collected spine sizes across eight time points (at −15, −10, −5, 2, 10, 20, 30, 40 min, where the negative numbers refer to the pre-stimulation) to study the spine dynamics. This data set consisted of three baseline observations, followed by glutamate uncaging or sham-uncaging, followed by another five time points. This allowed us to directly observe the effects of the LTP induction on spine populations and incorporate how the newly potentiated synapses and their unstimulated neighbors evolve within a single model. For information on the size of the datasets of this study, see Table 1.

We estimated the synaptic strength at each time point by measuring the size of the spine head[32,35,36] since many synaptic parameters correlate with head volume[37,38]. To this end, we biolistically overexpressed GFP in single neurons and imaged short stretches of dendrite over time. We show an example image, including semi-automatically generated ROIs used for measuring spine head size in Fig. 1a. We have highlighted a synapse with a gray rectangle in Fig. 1a and depicted its different sizes at different time points in Fig. 1b to emphasize the variable dynamics spines undergo. These recordings are performed in an imaging solution containing tetrodotoxin (TTX), picrotoxin, and with nominally 0 mM $Mg^{2+}$. Under these conditions, in the absence of neuronal spiking and experimentally imposed stimulation, spines constantly fluctuate spontaneously in size over time.

However, despite this variability, the distribution of spine sizes (Fig. 1c) is remarkably stable over time. Its shape is right skewed and exhibits a long right tail, in line with results reported previously across a variety of experimental studies[15,24]. Notably, we observed that the mean of the spine population is also remarkably stable, in contrast to the dynamics of the individual spines (see inset of Fig. 1c). We note that the distributions of spine size

**Table 1 Details of the activity-dependent and activity-independent experiments.**

| Experiment | # of animals | # of slices | Total # of spines | # of homosynaptic spines |
|---|---|---|---|---|
| Activity independent (no stimulation) | 21 | 47 | 830 | N/A |
| Activity dependent (7 spine stim.) | 5 | 10 | 204 | 65 |
| Activity dependent (15 spine stim.) | 6 | 15 | 338 | 187 |

The table shows the number of animals, slices, and spines analyzed in each experimental condition, as well as the total number of spines and the number of homosynaptic spines (i.e., stimulated spines) in response to the 7 or 15 spine stimulation experiments.

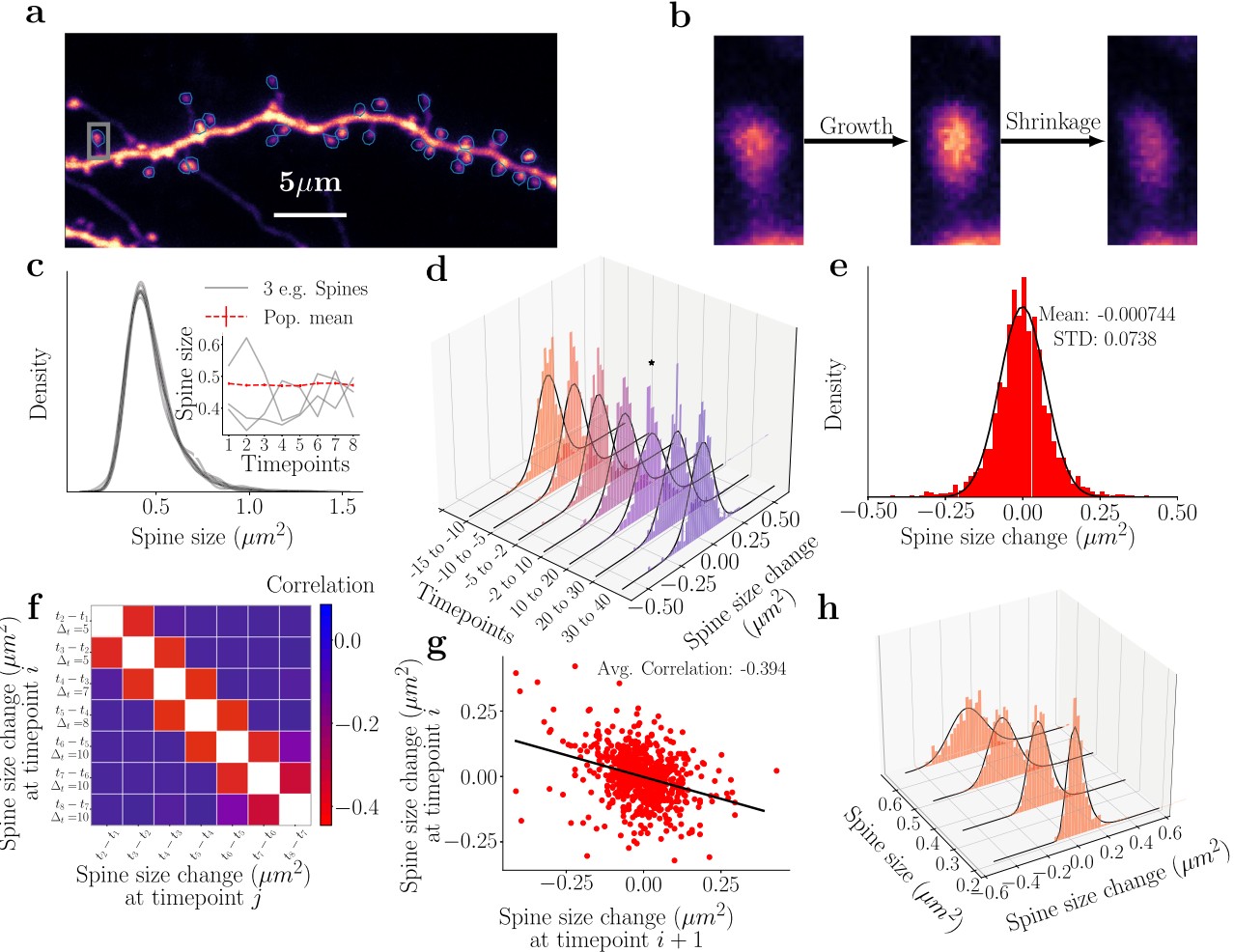

**Fig. 1 Experimentally measured population dynamics of activity-independent spine turn-over. a** An example of a GFP-expressing CA1 neuron whose spine dynamics we analyze and model. **b** Example of spontaneous dynamics at the single spine level. The spine (marked by a gray rectangle in (**a**) exhibits both growth and shrinkage in the observed time frame. **c** The spine sizes follow a temporally stable right-skewed distribution with a long tail. Each gray line refers to a different snapshot distribution, which shows significant overlap. Inset: The mean size of the full spine population (red) is shown across time along with the dynamics of selected spines (gray) at each time point, where the time points are at −15, −10, −5, 2, 10, 20, 30 and 40 min. **d** Collective distributions of the spine size changes (Δs) from time point to time point follow a Gaussian distribution. The black lines denote the corresponding Gaussian fits. The * denotes the single distribution that is significantly different ($p < 0.05$ when tested with KS test). Another depiction of these changes, which highlights the difference in the distribution is seen in Supplementary Fig. 2. **e** The collection of all spine changes across all time points follows a zero mean Gaussian distribution and a standard deviation of ≈0.074. **f** Spine sizes display correlations across time, whereby the neighboring time points are negatively correlated (negative off-diagonal values). **g** Correlation of two time points. **h** Evaluating spine size changes as a function of the spine size across time points shows that small spines exhibit a narrow distribution of spine size changes while larger spines show larger variability, black lines represent the corresponding log-normal (with no statistical difference seen between the dataset and a log-normal distribution) fits of the data.

changes (Fig. 1d and Supplementary Fig. 2) exhibit Gaussian behavior with no significant difference between time points (Kolmogorov–Smirnov (KS) test not significant except for the change from +10 to +20 min, which is marked by an *). We can also collect all these changes into one distribution and estimate the sample mean, $\mu$, and sample standard deviation, $\sigma$. The resulting distribution and sample statistics are seen in Fig. 1e. The spine size changes are robustly negatively correlated between neighboring time steps (see darker red colors in Fig. 1f). This effect is on the scale of 10s of minutes in our data, which is much shorter than the day-long spine correlations (which also have smaller values) reported by previous studies[12,30]. We note that this correlation also persists over the entirety of our experiments, as long as the timesteps immediately follow each other, e.g., computing the correlation of timestep 4–timestep 2 and timestep 6–timestep 4 (see Supplementary Fig. 3i). Finally, differently

sized spines exhibit different spine change distributions (see Fig. 1h) (KS test performed between samples led to $p$ values all under 0.05), which are all well-described by log-normal distributions (black lines).

These experimentally observed results in our data lead us to the following question: **given the dynamics of the individual spines (oscillatory, small vs big), how is the steady size distribution maintained?** We answer this question by introducing an abstract stochastic model that includes the lowest number of model parameters to maintain model tractability such that it captures the following key features of our experimental data:

1. The temporal spine dynamics need to remain stable around the distribution observed in the dataset (Fig. 1c). As a consequence, the mean of the distribution needs to remain stable through time (Fig. 1c—inset).

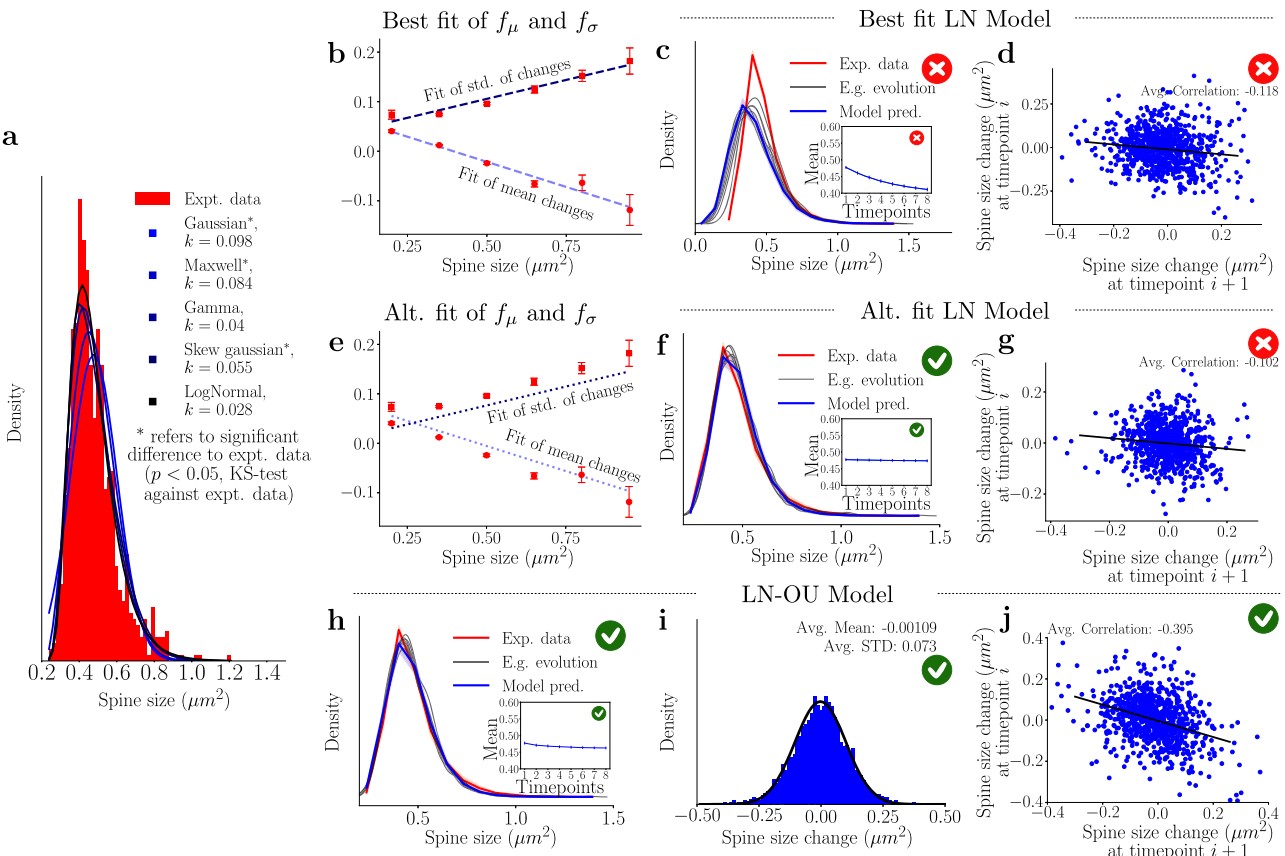

**Fig. 2 Utilizing the spine size dependencies to define the log-normal models.** Red crosses denote when the plotted model violates experimental observations, while green ticks indicate agreement with experimental data. **a** Fitting of different distributions to the spine size distribution, with *k*-values from the Kolmogorov–Smirnov test that show the best fit. The log-normal distribution best fits the spine size distribution. **b** Sample means and standard deviations of activity-independent plasticity for different subsets of spines can be used to obtain a linear fit between spine size and mean and standard deviation of their future size changes. Here the error bars represent the 95% confidence interval of the statistics after bootstrapping. **c** Simulations using the linear fits from **b** do not result in a stable distribution. E.g., *evolution* refers to one example simulation of spine sizes. The inset represents the simulated mean, which decreases significantly. **d** The correlation obtained from one example step of the best fits log-normal simulations. The value of the slope is ≈0.1, which is smaller than the correlations required. **e** Altered linear fits are used to achieve modeling goals. **f** Distribution obtained from the simulation when the altered linear fits of the sample mean and standard deviation are used. The stability of the distribution is achieved as well as that of the mean (inset). **g** The correlation obtained from one example step of the altered fits log-normal simulations. The value of the slope is ≈0.1, which is smaller than the correlations required. **h** The distribution obtained from using the best linear fits (*b*) for the LN-OU (Eq. (4)). Significant stability is observed (the inset represents the mean of the simulations). **i** Simulated activity-independent plasticity of the interpolated LN-OU model, showing clear Gaussian properties. **j** The correlation of the LN-OU process demonstrates a significantly more negative correlation in line with the desired model goals.

2. If we start at another distribution, e.g., a uniform or delta distribution, the model should return to the original stable distribution. This assumption does not arise directly from the observed data but more from the fact that as neurons grow and change, the initial spines could start small and still end up at the distribution of (Fig. 1c), which is stable over the timescales we consider (approximately 10 min). Therefore, to retain biological realism, we will include this feature.

3. The dynamics of spine changes and their distribution from one time point to another should follow a Gaussian distribution (Fig. 1d, e).

4. Time points immediately following each other should be negatively correlated with each other (Fig. 1f, g). This negative correlation suggests an oscillatory dynamic component.

**The Log-normal based model.** To understand the necessary noise profile driving spine size changes, we start with a more in-depth observation of the experimental data. The overall distribution of spine size changes over time appears to be Gaussian (Fig. 1e), which may imply a model that is based on Gaussian dynamics. However, when we attempted such a model, we observed that there were fundamental problems that did not reconcile with the experimental results (see "Methods" and Supplementary Fig. 4). In fact, we note that the overall profile of the spine size population is a skewed, log-normal-like profile (Fig. 2a). Furthermore, when we consider the changes in spines with different initial values separately, the distribution of changes also exhibits a skewed profile (Fig. 1h). Moreover, we note that these distributions differ from each other, suggesting that spines belonging to different size intervals behave in a fundamentally different way. Thus, we introduce a model with a noise profile, $\eta_i$, which is sampled from a set of log-normal distributions to define the dynamics of the spine size at time $i$, $V_i$, as

$$V_{i+1} = V_i + \eta_i, \eta_i \sim \text{Lognormal}(\mu_{log}(V_i), \sigma_{log}(V_i), -\hat{\delta}), \quad (1)$$

where $\mu_{log}$ and $\sigma_{log}$ are parameters that depend on the spine size $V_i$ and determine the shape of the log-normal sampling distribution and $\hat{\delta}$ is a shift parameter (see "Methods" for more

detail). To determine the dependence of $\mu_{log}$ and $\sigma_{log}$ on the size of the spine $V_i$ we assume, following observations seen in Yasumatsu et al.[12,15,24], that there exist two linear functions $f_\mu$ and $f_\sigma$ that map spine sizes onto the corresponding log-normal change parameters. However, rather than finding the linear functions that are optimal for all spines which (i) becomes computationally expensive, (ii) can lead to overfitting, or (iii) leads to difficulty inferring the underlying distribution due to insufficient data, we simplify the above model by binning spines in equal-size bins and then evaluating the sample means and standard deviations of those bins. This provides exactly the linear functions $f_\mu$ and $f_\sigma$ which allows us to estimate the sample means and deviations for all spine sizes (denoted by $\cdot_s$), i.e.,

$$\mu_s(V) = f_\mu(V) \qquad (2)$$

$$\sigma_s(V) = f_\sigma(V), \qquad (3)$$

These values can be used to estimate the parameters of the underlying normal distribution, which can then be transformed into the parameters to define that log-normal distribution ($\mu_{log}$ and $\sigma_{log}$) using Eqs. (12) and (13) and that we use to generate our noise profile. We note that previous work (including that of Hazan and Ziv[24]) found linear relations between the spine size squared and the variance and mean. We saw that such fits were equally effective as the fits presented here, and lead to similar results (see Supplementary Fig. 5). The fits for $f_\mu$ and $f_\sigma$ can be seen in Fig. 2b and lead to the following interesting results: (i) small spines have a positive mean change and have smaller standard deviation, so they tend to grow but are less variable and (ii) large spines have a negative mean change and larger standard deviation, so they tend to shrink and are more variable. We can use these insights to generate the first model, which we call the *Best fit LN Model* (LN for Log-normal) in Fig. 2, and study the properties of arising size dynamics (Fig. 2c, d). The generated results, reported in Fig. 2c, do not recreate the desired experimental characteristics, i.e., the mean of the simulated distributions (inset of the same figure) decreases, and the negative correlation is too small (compare Fig. 2d and Fig. 1g).

We notice, however, one crucial fact: by slightly altering the "best" linear fits of the means and standard deviations (raising the mean and lowering the standard deviation—see Fig. 2e), we obtain a new model (*Alt. Fit LN Model*) and excellent agreement with the experimental size distribution (Fig. 2f), still, however, underestimating significantly the correlation between subsequent changes (Fig. 2g). We can alleviate this by implementing the negative momentum term (see Eq. (9) in the "Methods") and using the altered fits (see Supplementary Fig. 3a, b, where we replicate the size distribution and the negative correlation). Despite the excellent agreement with the experimental results, we found it necessary to use the manually tuned fits for obtaining the mean and the standard deviation. As such, when implementing the Alt. Fit LN model, we were not using the optimal fits shown in Fig. 2b. We assume that the discrepancy in using the optimal fits is not due to any noise arising from the experimental set-up but, instead, because we are missing a crucial facet which the "altered" fits are accounting for. These observations lead us to introduce two key modifications in model (1):

1. to recover the negative correlation between subsequent size changes, we introduce the negative momentum term (also introduced in the "Methods" section (Eq. (9)) and Supplementary Fig. 4d, e);
2. by noticing that the manual changes applied to the fits are equal across all spine bins (Fig. 2e), we propose that an additional global drift term can recover the experimentally reported dynamics of the spine while allowing the

differential analysis of spine dynamics in different size groups. Therefore, we also add a global OU drift term (referred to as *Drift* below).

The parameters of our final model, referred to as the Lognormal–Ornstein–Uhlenbeck model, or *LN-OU model*,

$$V_{i+1} = V_i + \underbrace{\text{Lognormal}(\mu_{\log}(V_i), \sigma_{\log}(V_i), -\hat{\delta})}_{\text{Long-term stochasticity}} - \underbrace{\tilde{\theta}(V_i - \tilde{\mu})}_{\text{Drift}} - \underbrace{\theta(V_i - V_{i-1})}_{\text{Negative momentum}}$$

$$(4)$$

are fitted to achieve the best match to the experimental data. The resulting simulation is illustrated in Fig. 2h–j and indicates that we correctly reproduced all the experimental data we started out with in Fig. 1. Both the size distribution and the collective size change distributions are captured accurately and maintain a correct degree of negative correlation between subsequent size changes.

In summary, we have introduced a combination of two simple log-normal models that satisfy all our modeling requirements (see Fig. 2e–j for conditions 1, 3, and 4 and Supplementary Fig. 3e–h for condition 2). Constructed with the linear relations between spine size and mean and standard deviation of subsequent changes in mind, the model satisfies all modeling conditions we had set ourselves. Furthermore, this model introduces a slow-time scale (long-term stochasticity and drift) as well as a fast-time scale (negative momentum) that allows us to gain insight into the underlying processes of activity-dependent plasticity. For plausible links to biological mechanisms, see the Supplementary Material where we discuss possible links to actin dynamics[39,40] and geometric brownian motion[41]. Finally, this model is simple to implement and provides insights into the process that possibly underlies activity-independent plasticity.

**How LTP alters the spine size distribution**. Previously, all spines along the imaged dendritic branch were combined into one set, as there was no obvious manner to differentiate them (apart from their initial size). However, as we deliberately elicited plasticity by uncaging glutamate at a group of spines, we can now introduce two distinct spine sets: those that have been stimulated (homosynaptic, i.e., those synaptic targets which have specifically been targeted for sLTP) and those that are left untouched (heterosynaptic, i.e., spines on the same dendritic stretch that are not directly potentiated). We emphasize that the heterosynaptic spines, which were not targeted by the laser for glutamate uncaging despite sharing the same dendritic branch as the homosynaptic spines, are distinct from the spines from the previous sham stimulation spines, which were targeted by the laser, but due to the omission of glutamate did not undergo potentiation. We restrict the heterosynaptic spines to be within 4 μm of the stimulation sites and treat them as one distinct group. Finally, to have a sufficient number of homosynaptic spines we chose to stimulate 15 distinct spines sharing the same dendritic branch. Before we apply the previously defined log-normal model to this data set, we will need to understand the effects of stimulation on activity-independent spine turnover.

Beginning with the collective spine distribution Fig. 3a, we note that the pre-stimulation (red) and post-stimulation (blue) stationary distributions are significantly different. This is also reflected in the set of time point means (top inset). This implies that the spine dynamics before and after stimulation can be classified as activity-independent plasticity around the respective stable distribution but that the act of spine stimulation acts instantaneously (at our time resolution) shifting the distribution of spine sizes. To quantify the distributional change further, we measured the amount of information or "uncertainty" within the given spine size distributions[42]. Hereby, we use Shannon entropy,

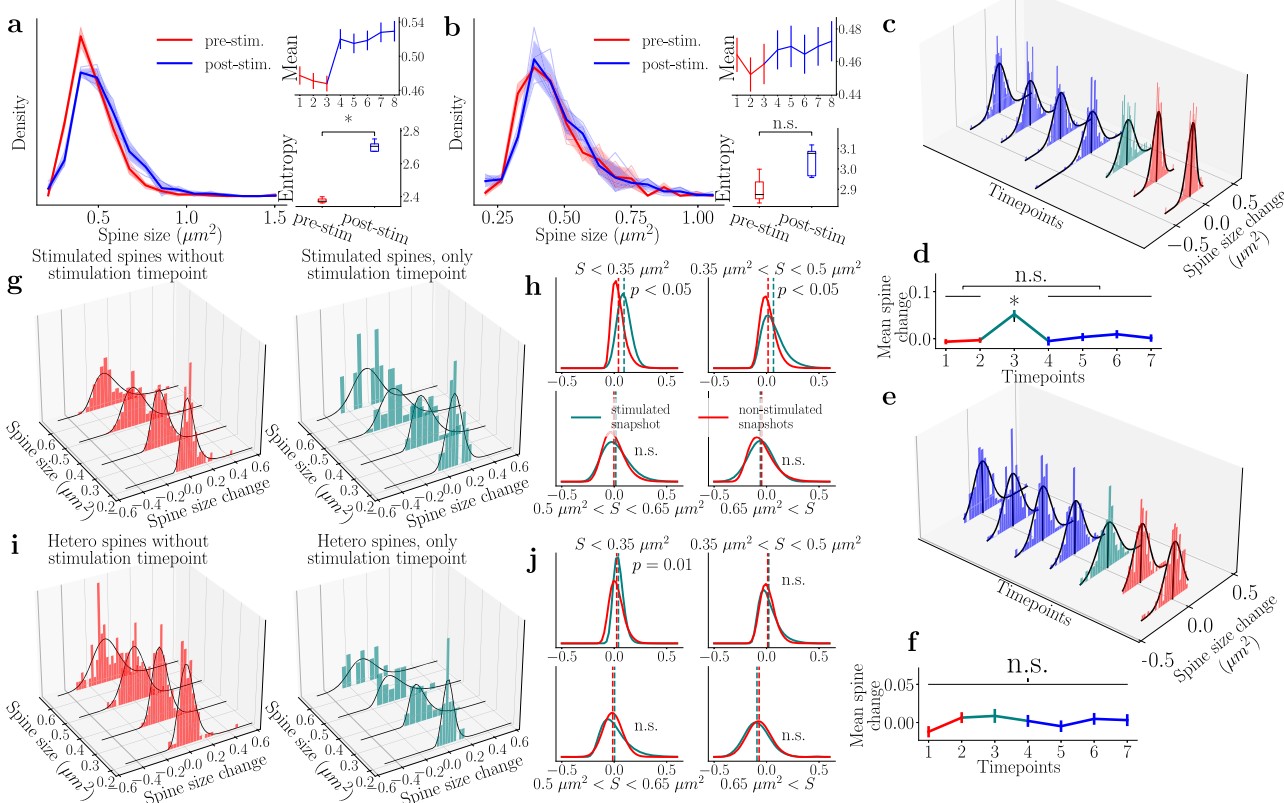

**Fig. 3 Stimulation of spines leads to a distinct shift of the spine size distribution that is mainly driven by growing small spines. a, b** Homosynaptic and heterosynaptic spine size distribution at different time points, with red and blue referring to pre-stimulation and post-stimulation, respectively. Sample mean and entropy are shown. * and n.s. refer to $p < 0.05$ and $p > 0.05$ of a two-sided $t$ test comparing pre- and post-stimulation. **c** The collective change dynamics of all homosynaptic spine sizes follow a Gaussian distribution. Teal represents the spine size change directly after the stimulation. **d** The mean of spine change from time point to time point computed for all homosynaptic spines together. A one-way ANOVA test reveals that only the stimulation time point is significantly different. All other time points are not significantly different from activity-independent fluctuations. **e** Distribution dynamics of heterosynaptic spines time point to time point follows a Gaussian distribution. **f** Temporal change in the mean of spine changes in the heterosynaptic spines. A one-way ANOVA test reveals lack of statistical differences across time. **g** Splitting up the size changes in homosynaptic spines according to their initial size reveals a large difference in activity-independent plasticity distributions. The left figure represents all the time points without stimulation, and the right is the single time point immediately after stimulation. The associated black lines represent log-normal fits to the data. **h** A comparison between the log-normal fits for the size buckets reveal the effects that the stimulation has on the different spine sizes of the homosynaptic spines. Red refers to the non-stimulated time point, and the teal to the stimulated ones. The $p$ value in the figure refers to a KS test performed on the data in (**g**) to verify whether the samples come from different distributions. **i, j** Same procedure as **g, h** but for the heterosynaptic spines.

which quantifies how many *bits* a distribution has and is defined by

$$H(X) := -\sum_{x \in \mathcal{X}} p(x) \log p(x) = \mathbb{E}[-\log p(X)] \qquad (5)$$

where $\mathcal{X}$ defines the full set of possible sizes and $p(x)$ is the normalized distribution of different spine sizes. We calculate the information capacity of the spine size distributions before and after stimulation. We note that the lower inset of Fig. 3a shows a significant increase in the potential information-retaining capacity of the neuron due to the stimulation. In contrast, Fig. 3b shows that the heterosynaptic spine (<4 μm) size distributions and sample means do not show any significant shift during stimulation. The entropy does increase, albeit not significantly.

The changes from time point to time point of both the homosynaptic (Fig. 3c) and heterosynaptic spines (Fig. 3e) mirror those of the activity-independent plasticity (Gaussian distributions). For Fig. 3c, we see that stimulation protocol (depicted in teal) induces a significant shift in the location of the change distribution (see Fig. 3d, which depicts the mean change) but no significant change in the shape. In contrast, the heterosynaptic spines do not exhibit a significant difference in shape or location

from the other time points (see Fig. 3f). Furthermore, when considering the averages of the changes at each time point (vertical black lines in the 3D plot and dynamic plots immediately below), we see that the stimulation time point for the homosynaptic spines is significantly elevated over the other time points. This elevation supports the "shift" event we observed in Fig. 3a. We also note no significant difference between all other time points. Given that we assume that the pre-stimulation time points are akin to activity-independent plasticity (i.e., there is no knowledge that a stimulation event is about to occur), we can then assume that the distribution of the spine changes after stimulation is also defined by activity-independent plasticity. The heterosynaptic spines do not demonstrate such an elevation, and so we assume that, for the most part, these spines undergo activity-independent plasticity.

We next divided the population of spines according to their sizes in bins of $0.15\,\mu m^2$, which can be seen in Fig. 3g (homosynaptic spines) and Fig. 3i (heterosynaptic spines). As we assume that all non-stimulation time points represent activity-independent plasticity, we collect these into one and plot these changes in spine size in the left figures. The figures on the right

only show the time point immediately post-stimulation. We note that these all are approximated by log-normal distributions (fits in black) (cf. Fig. 1h). We can also compare the distributions of each bin (Fig. 3h, j—homosynaptic and heterosynaptic spines, respectively). The inset *p* values refer to a KS test between the two data sets. Differences were significant for homosynaptic spines only under 0.5 μm², and for the heterosynaptic spines, only under 0.35 μm². This suggests, in line with results seen in Matsuzaki et al.[32], that small spines are proportionally more affected by the glutamate uncaging event and play a more important role during the acquisition of new memories. In contrast, larger spines are more stable and do not change significantly from the baseline activity-independent plasticity. Finally, we observe that the stimulated spine change distribution is narrower for the small (<0.35 μm²) heterosynaptic spines (Fig. 3j, teal vs red). This narrowing appears skewed to the right, such that the decrease in activity-independent fluctuations could be preferentially associated with the shrinkage of the small spines. In contrast to the stimulated small spines that undergo growth, neighboring small spines experience the stimulation only peripherally. In such a case, the components that induce growth may not reach levels sufficient to actually cause growth while they may be present at levels that could still counter (or compete with) activity-independent shrinkage.

**The LN-OU model applied to stimulated spines**. To apply our model to the stimulation scenario, we need to determine the new linear dependencies on spine size and log-normal statistics that arise. As a first step, we analyze the sample means and standard deviations for the homosynaptic (Fig. 4a) and heterosynaptic spines (Fig. 4b) while omitting the stimulation snapshot. We note that the resulting model agrees well with the previous fits (in gray), confirming our observation that the pre-stimulation baseline model applies.

We next study the stimulation snapshot and observe that the model fits for the heterosynaptic spines in Fig. 4c reveal only a slight deviation in the smallest spines from the activity-independent baseline. Therefore, for simplicity, we consider that the heterosynaptic spines undergo activity-independent baseline dynamics at all time points. For the homosynaptic spines in Fig. 4d, a different behavior emerges. We see that the standard deviation is tilted upwards, meaning that the resulting log-normal distribution has increased its standard deviation and that the spines became more variable during stimulation. We note that this increase follows intuitively for the following reasons; as the spines are rapidly enlarged by the potentiation protocol, their variance will also be increased because (i) they have grown beyond the normal size of activity-independent plasticity and (ii) they are now large spines, which have been demonstrated to have larger variance compared to small spines. This increase in the standard deviation is only observed in the medium-sized bins and not for the small or large spines. This could be explained by the fact that the medium spines, which are able to grow to be the size of large spines, now exhibit the characteristics of those large spines, including an increased variance. Additionally, in this current study, medium-sized spines were preferentially chosen for stimulation, as previous studies have shown that this population are most labile in terms of potentiation (for example Matsuzaki et al.[32]). It is possible that had we chosen to selectively target groups of large mushroom spines, or alternatively filopodia, that the outcome could be different. Finally, the mean spine change exhibits a distinct linear trend, i.e., the smaller the spine, the larger the mean increment compared to the fit from the activity-independent plasticity.

To understand which parameters of the model need to be altered to replicate the stimulation time point for the homo-synaptic spines (Fig. 3a), we will alter each component, long-term stochasticity and drift, of the log-normal model, individually. Additionally, we assume that the negative momentum term is a term that is inherent to activity-independent plasticity, i.e., it occurs as a stabilization mechanism and counters the previous stochastic change. As stimulation is a directed activity, negative momentum would hinder the growth of spines after stimulation by promoting shrinkage and imply that the previous stochastic activity-independent plasticity directly affects the subsequent activity-dependent change. Consequently, we choose to deactivate this term in the model during the stimulation step to avoid this scenario. However, future studies could consider including this or a generalized negative momentum term and study its role for the resulting synaptic size distribution.

First, we changed the long-term stochasticity component of the model by using the linear fits for the stimulation time point (Fig. 4e). The fast component of the stimulation is reproduced; however, by keeping the drift constant, we slowly return to the original distribution. This is not what we observe in our experiment with the stimulation of 15 spines (Fig. 3a). We note that we do observe the decay back to baseline for a separate case in which only seven spines were stimulated (Supplementary Fig. 6a). From this, we can assume that the sustained LTP response is linked to the higher drift term and implies that the long-term stochasticity component replicates the immediate potentiation while the drift portion leads to the sustained effect. Further evidence for this assumption can be seen in Fig. 4f, where only the drift term is altered at all points after stimulation, and the linear fits are taken from the activity-independent plasticity. The change in the mean and distribution is slower and does not include instantaneous potentiation. Previously, the long-term stochasticity and drift components were active on similar time scales. For the stimulation we see that the long-term stochasticity enacts instantaneous changes to the structure of the spines over the timescale we considered, while the drift towards the new steady state occurs afterwards on a longer time scale.

Finally, we alter both components by changing linear fits at the time point post-stimulation and the drift parameter *μ* after stimulation. Fig. 4g demonstrates that we reproduce a distinct set of stable distributions before and after stimulation on the required timescales (cf. Fig. 3a). Thus, the LN-OU model reproduces the experimental results of both types of plasticity. To achieve the jump in distributions seen in the simulations, the full linear fits seen in Fig. 4d were used for the long-term stochasticity. Furthermore, given the observation that small spines are most affected by stimulation, we examined the effect of only changing the parameters of the smallest spines in the model (Fig. 4h). We, therefore, only increased the sample mean of the spines with an initial area of <0.35 μm² during the stimulation and treated the stochastic component of all other spines as if they were undergoing activity-independent plasticity. The drift parameters were applied as above, as they affect all spines equally. In other words, we were altering the slow component of all spines but only altering the fast potentiating component of the smallest spines. With this change, we can reproduce the experimental results with no noticeable difference from when we used the full linear fits (compare Fig. 4h and Fig. 4g).

We also calculated the Shannon entropy of the simulated distributions[42–44]. The result of this calculation can be seen in Fig. 4i. In all cases, we significantly increase the information encoding capabilities of the synaptic weight distribution after stimulation. However, only changing the long-term stochasticity

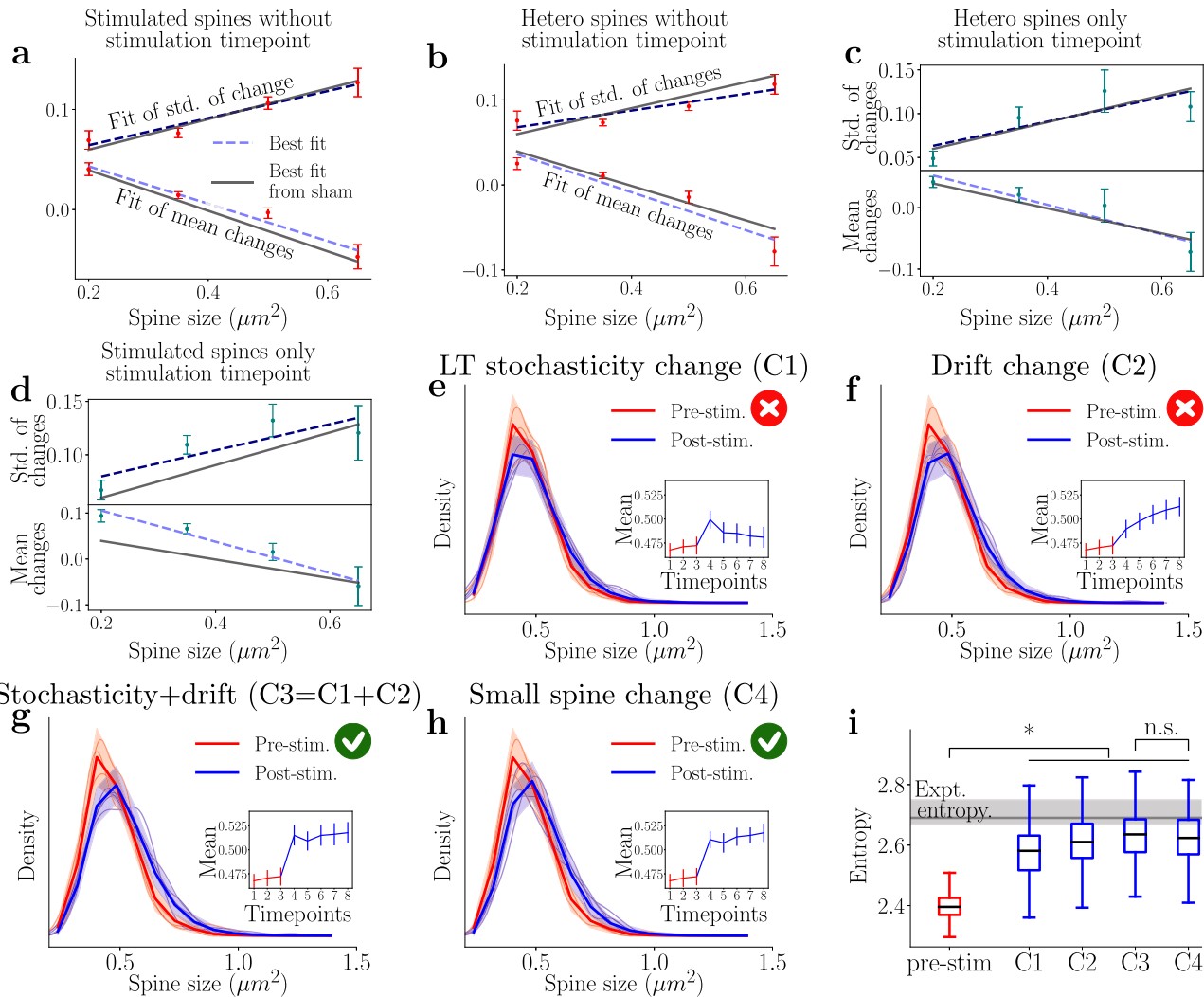

**Fig. 4 The Log-normal–Ornstein–Uhlenbeck (LN-OU) model can reproduce homosynaptic spines dynamics even if only small spines are altered.** The figure shows how the model can reproduce the dynamics of homosynaptic spines by changing the behavior of small spines only. The red crosses denote when the model violates experimental observation, and the green tick denotes agreement with the data. **a**–**d** Subsets of homo- and heterosynaptic spines were split according to size, and linear fits were carried out for the sample mean and standard deviation of the spine activity. Here the error bars represent the 95% confidence interval of the statistics after bootstrapping. **a**–**c** Fit of the non-stimulation snapshot of the homosynaptic spine and all snapshots of the heterosynaptic spines show good agreement with the activity-independent plasticity fits (gray). **d** Stimulation snapshot of the homosynaptic spine shows a difference in the fit for smaller spines. **e** Model simulation dynamics pre- and post-stimulation. The immediate growth is observed but not sustained when only changing the stochastic portion. **f** Simulation results when the long-time stochasticity was kept the same as the model in Fig. 2e, and only $\tilde{\mu}$ was changed to reflect a new stable point. **g** Represents the simulation results when the two previous changes are implemented in tandem, mirroring the sustained LTP seen in Fig. 3a. **h** A simpler change in the stimulation model is introduced, where $\tilde{\mu}$ is changed as in the previous figures while the long-time stochasticity are only shifted for the spines <0.35 μm² in size. **i** The Shannon entropy of the simulated distributions is calculated and compared to the experimental value. The stimulation event adds significant information in all cases, and there is no significant difference when the fast change is only applied to small spines. Center lines of the whisker-plots refer to the median simulated entropy and whiskers to the inter-quartile range. * and n.s. refer to $p < 0.05$ and $p > 0.05$ of a two-sided $t$ test comparing each of the different datasets with each other.

(i.e. the short-time effect of the stimulation) leads to a smaller increase in entropy which could then conceivably decay back to pre-stimulation levels after the observed time period. The other alterations to the model to emulate the stimulation event have markedly higher entropy values (akin to those observed experimentally). Notably, there is no significant difference in the entropy of the fully altered and small spine models. We conclude that any model that aims to reproduce the population dynamics of spine sizes can focus on the smallest spines to simplify the simulation process while still achieving good results, at least over the time scales considered in this study.

## Discussion

In the present study, we considered experimentally recorded population dynamics of both stimulated and unstimulated spines sharing the same dendrite. Inspired by previous work[12,15,24], we have introduced a model framework incorporating the dynamics for spontaneous and plasticity-driven spine changes we measured in our data. Specifically, we observed a stable right-skewed distribution of spine sizes in which the dynamics of small and large spines seemingly follow different computational rules.

We built a model operating at the level of synaptic populations that can be represented by a single stochastic differential equation

and refrained from modeling detailed molecular principles, such as those reported in Shomar et al.[45] or Bonilla-Quintana et al.[30] following the historic footsteps of previous abstract models[12,15,22]. Taking this high-level view, we gained insights into activity-independent plasticity and incorporated different time scales. Previous studies (e.g., refs. [12,15,24,45]) have offered a rich set of stochastic modeling options to describe specific individual effects present in the spine dynamics. In accordance with previous literature, we followed the ergodic hypothesis for our modeling. However, due to the 55-min recording window in our data set, we could not test ergodicity directly or show that each spine explores the full phase space (see Supplementary Fig. 3c, d).

Our model incorporates a fast and a slow mechanism that both have implications for the synaptic stability of a neuron. The fast spine changes that are anti-correlated with previous size changes may prevent a winner-take-all system by differentiating between small and large spines. Enforcing that large spines shrink on average guarantees that spine size remains bounded and is in line with the long tail of the spine size distribution. Additionally, in our experiments we observed that small spines preferentially showed a positive size change (Fig. 3h), and therefore they could act as points of information acquisition during plasticity induction. In contrast, large spines did not change their dynamics significantly after the stimulation, such that the large spines could help maintain the stability of previous state. In the absence of direct plasticity cues, we observed that large spines were more variable and, on average, prone to shrinkage. Large spines, with their intricate structural complexity, require a larger number of proteins, membrane traffic, and actin filaments to support their maintenance, leading to a higher energy cost. This would justify favoring size reduction for large spines in line with an energy-efficient (homeostatic) system that degrades preferentially large spines (older memories that became obsolete) to optimize storage and energy in the brain. We note that our results do not explain how a small subset of spines (e.g., large spines representing selected memories) can be preserved over timescales of days or months (for a brief discussion on how longer timescales could be incorporated in the model, see the numerical methods section).

Our model builds on and extends several modeling studies addressing the differences in the dynamics of small and large spines. An early study by Yasumatsu et al.[12] split small and large spines into different categories based on manual group assignment to model activity-independent plasticity. Our work proposes a plausible mechanism for activity-independent plasticity that avoids such rigid categories. Another study by Loewenstein et al.[15] found that the temporal changes in spine size could be approximated by a model incorporating two timescales by using multiplicative dynamics and Ornstein-Uhlenbeck processes, consistent with the fast and slow components of our model. A different model by Statman et al.[22,24] used the Kesten process to describe synaptic remodeling dynamics. Shomar et al.[45] introduced a molecular model that explained how size fluctuations and distributional shapes can emerge from stochastic assimilation and removal of synaptic molecules at synaptic sites. Finally, Bonilla-Quintana et al.[30,31] used actin dynamics to model rapid, spontaneous shape fluctuations of dendritic spines, predicting that these polymerization dynamics self-organize into a critical state that generates negative correlations in spine dynamics on short time scales.

Additionally, a vital aspect of our study is the consideration of both stimulation and activity-independent plasticity in a single experimental paradigm and single mathematical model. Previous imaging studies have either limited glutamate-uncaging to single spines[32,46], or small clusters of spines[47,48] and did not monitor population-level changes in synaptic sizes. Others monitored multiple spines while applying global chemicals to induce

plasticity, e.g., ref. [49]. Here, we could confirm one of the results of Matsuzaki et al.[32] that small spines are the prime targets for growth and, therefore, may be the substrates for the acquisition of new memories and, consequently, that large spines are likely to be the reservoirs for long-term memories[33,34,50]. Model justifications for distinct dynamics in small and large spines has been discussed in Shouval[51] that proposed a mechanism based on clusters of interacting receptors in the synaptic membrane, Bell et al.[52] who considered a reaction-diffusion model of calcium dynamics and Jozsa et al.[53] that showed that discrete, stochastic reactions and macroscopic reactions could be exploited for size-dependent regulation. Interestingly, we observed that the distribution of spine sizes was different post compared to pre-stimulation. In contrast, we saw that the changes in spine size ($\delta V$), when viewed as a population across all time points (longer than 2 min away from plasticity induction), were indistinguishable from activity-independent, spontaneous changes.

Thus, our model provides a common stochastic framework that helps understand spine plasticity operating spontaneously after stimulation. Finally, we considered the entropy and information content of the synaptic populations. Entropy is a measure of disorder in a system and can be measured by observing the diversity of synaptic sizes in a neural network. Higher entropy implies a more disordered system that allows for more diverse information encoding capabilities. Following LTP stimulation, we observed an increase in the range of synaptic sizes and, thus, a larger set of possible states consistent with higher entropy. This higher entropy could facilitate learning by enabling the network to differentially encode a wider range of inputs. Secondly, entropy can also reflect the stability and robustness of synaptic connections. A higher entropy, reflected by a more diverse distribution of synaptic strengths, could make a network less sensitive to changes in individual synapses. This increased ability to buffer against noise or disruptions, such as the loss or weakening of specific synapses, helps promote the overall robustness of the network.

Our study provides not only a common framework for understanding spontaneous versus evoked dynamics across spines but also helps establish a unified view of various features related to spontaneous spine dynamics that align with prior reports obtained in different experimental preparations. Since spontaneous spine dynamics is often studied across both in vivo and in vitro preparations, slices, hippocampal or organotypic cultures, and across different brain regions confirming or differentiating these reports within a common model framework is an ongoing challenge. While our experiments are conducted in slices, e.g., ref. [24] has taken advantage of primary culture models to image spines over hours to days while monitoring fluorescently tagged PSD components. Similarly, Yasumatsu et al.[12] worked in hippocampal slices and employed different blockers to silence neuronal activity while observing several spine dynamics features compatible with our model and data. Other studies, such as Loewenstein et al.[15], imaged dendritic spines in vivo in the auditory cortex, measuring populations of spines over days to weeks. During imaging sessions, the mice were lightly anesthetized, but activity at these synapses evolved spontaneously between sessions, leading to synaptic strength changes. Interestingly, despite these differences in experimental preparations, many reported features align with our experimental data, including the right-skewed spine distributions and size-dependent statistics consistent with our model.

In summary, our study established a link between activity-independent spontaneous spine dynamics and directed synaptic plasticity. Within this modeling framework, we were able to unite new and previously reported synaptic features such as stable

distribution of spine sizes,[12,15,22], higher variability of larger spines vs. small spines[12,13,15,24], the oscillatory behavior of the spines[12,30] and incorporate into the same model plasticity-induced dynamics. This framework can open avenues for interpreting specific experimentally reported synaptic changes relative to spontaneous activity and help constrain plasticity models operating at the circuit level.

## Methods

**Preparation of organotypic hippocampal slice culture**. Organotypic hippocampal slices were prepared as previously reported Stoppini et al.[54]. Briefly, the brains of postnatal day 6–7 Wistar rat pups (Nihon SLC) were removed, the hippocampi dissected out, and 350-μm-thick transverse slices were cut using a McIlwain tissue chopper (Mickle Laboratory Engineering Co. sLTD. and Cavey Laboratory Engineering Co. sLTD.). These slices were then placed on cell culture inserts (0.4 mm pore size, Merck Millipore) in a 6-well plate filled with culture media containing 50% Minimum Essential Medium (MEM, Thermo Fisher Scientific), 23% EBSS, 25% horse serum (from Thermo Fisher Scientific), and 36 mM glucose. The slices were maintained at 35°C and 5% CO2 and used for experiments between DIV16-18.

The slices were perfused with 1–2 ml/min of artificial cerebrospinal fluid (aCSF) containing (in mM) 125 NaCl, 2.5 KCl, 26 NaHCO$_3$, 1.25 NaH$_2$PO$_4$, 20 glucose, 2 CaCl$_2$, and 4 mM MNI-glutamate (Tocris). The aCSF was continually bubbled with 95% O$_2$, and 5% CO$_2$ and experiments were carried out at room temperature. All animal experiments were approved by the RIKEN Animal Experiments Committee and performed in accordance with the RIKEN rules and guidelines. Animal Experiment Plan Approval no. W2021-2-015(3).

**Transfection and imaging of CA1 pyramidal neurons**. Organotypic slices were transfected with a Helios gene gun, and used for experiments 48–96 h later. For structural plasticity experiments, gold particles coated with a plasmid encoding EGFP were used. 50 μg of EGFP plasmid was coated onto 20–30 mg of 1.6 μm gold particles. The neurons were imaged at 910 nm on a Zeiss 780 microscope, and all data was analyzed offline.

**Dendritic spine imaging and glutamate photolysis**. Neurons were selected for imaging if their gross morphology appeared healthy. Single dendrites were selected visually for imaging and stimulation. Dendrites were imaged for a brief period of time by collecting a series of Z stacks of the dendritic arbor at a resolution of 512 × 512 and 4 × digital zoom, with 4× averaging, resulting in a final image size of 33.7 μm. The Z step between each image in the stack was 0.5 μm. For the induction of plasticity, spines on the dendrites were stimulated by applying a train of 60 pulses of laser light (4 ms each) using custom-written software, and uncaging glutamate at a distance of 0.5 μm from the spine head. Medium-sized spines with a clear spine head within the field of view were preferentially targeted for stimulation. A 2-photon laser source (720 nm) was used for photolysis of MNI-glutamate, and the stimulation was repeated at a rate of 1 Hz. For groups of homosynaptic spines, laser pulses were delivered in a quasi-simultaneous fashion, in which the first spine receives a pulse of glutamate (4 ms) which is followed by a short delay (<3 ms) as the system moves the laser to the next spine. This is repeated for all spines in the stimulated cluster and repeated at 1 Hz. For sham-stimulation experiments, MNI-glutamate was omitted from the aCSF.

**Size of the data set**. See Table 1.

**Image analysis**. To obtain the areas of the individual spines, which can be seen as a proxy for the strength of that synapse[55,56], were generated by using the area of an octagonally shaped ROI surrounding the spine head. The algorithm for the generation of this octagon is part of an in-house python code[57]. Briefly, the spine ROI was generated by using a semi-automatic in-house python package that took advantage of the inherent structures of the spines. The manual interaction involves a simple clicking on the interior of the spines while the ROI and subsequent measurement are performed automatically. Temporal shifting was corrected by using a phase cross-correlation algorithm implemented in SciPy[58]. Synapses that were partially obscured by the dendrite or overlapped with other spines were omitted from the analysis. All images shown and used for analysis are maximum-intensity projections of the 3D stacks.

**Statistical definitions**. Throughout this manuscript, we used the absolute change in spine areas, which is defined as follows:

$$\Delta V_i = V_i - V_{i-1} \qquad (6)$$

Error bars represent standard error of the mean, and significance was set at $p = 0.05$ (two-sided studentized bootstrap). To compare distributions against each other, the populations were taken (in the case where these samples were very large, randomly subsampled), and a KS test was performed. Single asterisks indicate $p < 0.05$. Fits of probability distribution functions were performed using SciPy. Correlations report the Pearson product-moment correlation coefficients. Unless reported otherwise, error bars in line plots refer to the standard error and in box-and-whisker plots refer to the inter-quartile range.

**Building a Gaussian model**. We start by considering the Gaussian distribution of the experimentally observed spine changes in Fig. 1d, e. Thus, a purely Gaussian model for the spine changes appears as a natural first choice. This model has the form:

$$V_{i+1} = V_i + \eta_i \qquad (7)$$

where $\eta_i \sim \mathcal{N}(\mu, \sigma)$ and $V_i$ is the size of a spine at time point $i$. While this model is simple and captures the experimentally observed statistics of spine changes, it exhibits an inherent incompatibility with other experimental results. Since a Gaussian distribution is naturally unbounded, this model permits infinitely large (negative and positive) spine size values.

Historically, the lack of bounds in a Gaussian distribution has been addressed via the introduction of bounding walls $W_l, W_r$, e.g., ref. [12]: at each time step, the value $V_{i+1}$ is reset to be within the range $[W_l, W_r]$, where $W_l < W_r$. This can be achieved, for example, by using either a bounce-back mechanism (i.e. a change in the opposite direction) or imposing no change, i.e., $V_{i+1} = V_i$. To investigate whether the introduction of walls can allow us to move forward with the Gaussian model, we implemented two walls ($W_l$ and $W_r$) which we set to equal the fifth percentile and the largest experimentally observed spine size, respectively. The resulting model simulations (using Eq. (7)) are seen in Supplementary Fig. 4a, where the dashed lines represent the walls. Despite a good agreement with the collective spine distribution, three conceptual issues rule out this model:

1. The left wall enforces a build-up of smaller sizes that leads to the desired asymmetry but also leads to a complete drop-off in spines smaller than this size (Supplementary Fig. 4a).
2. Spines are free to grow until they reach the right wall value, causing an overall increase in the population mean and a

biologically implausible growth at the right tail of the size distribution (Supplementary Fig. 4a, b).

3. The negative correlation between subsequent size changes is lost due to the memory-less additive Gaussian noise (Supplementary Fig. 4c).

Therefore, we will modify our model further to include a negative temporal correlation and achieve a biologically plausible spine size distribution. To this end, we will replace the purely diffusive process with an Ornstein–Uhlenbeck process. This approach was previously also used in Loewenstein et al.[15] to model activity-independent plasticity in a framework with multiplicative noise. Here we will be applying it in an additive manner:

$$V_{i+1} = V_i + \theta(\bar{\mu} - V_i) + \eta_i \qquad (8)$$

where $\theta, \bar{\mu}$ are the drift terms and $\eta_i$ is as above. We observe that this process, characterized by the deterministic drift towards the long-term average $\bar{\mu}$, can reproduce the experimental mean-reverting behavior if $\theta$ is large enough. However, if we choose $\theta$ to be too large, all the spine sizes will eventually stabilize in a narrow neighborhood around $\bar{\mu}$, which is inconsistent with the experimental observation that even after hours and days, there was a stable and diverse set of different spine sizes[12,17]. Adopting a set of different values of constant $\bar{\mu}$ for the different spines while keeping a high value of $\theta$ allows the recovery of this phenomenon but inevitably locks the spines each into their stable size and prevents them from changing from one size to the other. Therefore, to avoid these pitfalls, we introduce a drift $\bar{\mu}$ that is (i) unique to each spine and (ii) time-dependent. Thus we avoid both the global stable size as well as the local stable size. The simplest implementation of this principle is the introduction of a "negative-momentum" term, obtained by setting $\bar{\mu} = V_{i-1}$

$$V_{i+1} = V_i + \theta(V_{i-1} - V_i) + \eta_i \qquad (9)$$

This non-Markovian process contains a bounce-back mechanism that induces the spines that have grown in the previous step to have a higher probability of shrinking in the next one. Importantly, this effect vanishes at longer timescales. We implement this model by setting $\theta$ to achieve the experimentally observed correlation. The results of the simulations can overcome two of the three issues illustrated above: the population mean remains stable over time (Supplementary Fig. 4d, inset), and the oscillatory behavior reappears in agreement with the experimental observations (Supplementary Fig. 4e). However, the additive Gaussian term is still responsible for an improper tail-fattening and, ultimately, for an improper symmetrization of the spine size distribution. This fact and the observation that the different spine sizes exhibit different noise profiles (see Fig. 1h) show that more complicated noise-generating models are required to model activity-independent plasticity.

**The log-normal model.** In probability theory and statistics, the log-normal distribution is a continuous probability distribution of a random variable whose logarithm is normally distributed. That is, if the random variable $X$ is log-normally distributed, then $Y = \ln(X)$ is normally distributed. The log-normal distribution is parameterized by the mean, $\mu$, and standard deviation, $\sigma$, of the underlying normal distribution. The probability density function of the log-normal distribution is given by

$$p(x) = \frac{1}{x\sigma\sqrt{2\pi}} e^{\left(-\frac{(\ln(x)-\mu)^2}{2\sigma^2}\right)} \qquad (10)$$

where $x$ is the value of the log-normally distributed variable. As we will be modeling data that can take negative values (the spines can shrink) and the standard log-normal is only defined for

positive values, $x > 0$, we also need one additional parameter to characterize our distribution: the shift parameter. This parameter shifts the distribution so that $x > \delta$ where $\delta$ can be positive (shifted to the right) or negative (shifted to the left). The probability distribution is then

$$p(x) = \frac{1}{(x-\delta)\sigma\sqrt{2\pi}} \exp\left(-\frac{(\ln(x-\delta)-\mu)^2}{2\sigma^2}\right) \qquad (11)$$

Given access to the entire population of spine size changes, the parameters that define the log-normal distribution can be found by transforming the sample means and standard deviations ($\mu_s$ and $\sigma_s$) of the spine size changes as follows:

$$\mu_{\log} = \log\left(\frac{(\mu_s + \hat{\delta})^2}{\sqrt{\sigma_s^2 + (\mu_s + \hat{\delta})^2}}\right) \qquad (12)$$

$$\sigma_{\log} = \sqrt{\log\left(\left(\frac{\sigma_s}{(\mu_s + \hat{\delta})}\right)^2 + 1\right)} \qquad (13)$$

where we have introduced the positive term $\hat{\delta}$, which shifts the sample mean towards positive values. The choice of $\hat{\delta}$ is relatively trivial as long as all the values of the dataset are positive after the shift. This ensures that $\mu_{\log}$ is also positive, thus avoiding the log-normal distribution transformation accumulating values around $x = 0$. Once the parameters of the log-normal have been estimated, the model uses the log-normal distribution to generate the subsequent time points. This model then takes the form

$$V_{i+1} = V_i + \text{Lognormal}(\mu_{\log}, \sigma_{\log}, -\hat{\delta}) \qquad (14)$$

which mirrors the form of the original Gaussian model. Here, we include the $-\hat{\delta}$ to map our log-normal back to the original range of values that we observe in the data. We emphasize here that the change $V_i - V_{i-1}$ for each individual spine is log-normal but that the population change, i.e., the collection of all changes should still be normally distributed (c.f. Fig. 1e). By the central limit theorem and the assumption that the activity-independent plasticity of the spines is independent of each other, we will obtain this Gaussian nature as long as we have sufficiently many spines.

**Incorporating longer timescales in our model.** We briefly comment on the concept of "stability" used in this study. We recognize that our experimental timescale of tens of minutes is insufficient to definitely state that we are observing the population "steady-state." Indeed, the effects that lead to population changes could occur on timescales that far exceed our experimental timeline. Therefore, when we use the term "stable," we refer to the short-term effects rather than the possible longer relaxation times of the population dynamics.

Our experimental paradigm was limited to ~1 h. Therefore, the temporal components of our model are on this scale. Nevertheless, we can augment our model to study longer timescales and answer questions such as: is the shift to a "stable" distribution after stimulation truly stable over a long time horizon, or is there a possible decay that we cannot observe due to our shorter time paradigm?

We observed that altering only the long-term stochastic component of the log-normal OU model led to the shift to the new distribution and then decay back to the baseline (see Fig. 4e). We saw the stable post-stimulation size distribution only when the drift term was also increased. If we define the pre-stimulation drift term as $\tilde{\mu}_{pre}$ and the post-stimulation drift as $\tilde{\mu}_{post}$, then we

enacted the drift change as

$$\tilde{\mu}_{post} = \tilde{\mu}_{pre} + \Delta\mu \quad (15)$$

where $\Delta\mu$ is the increase in the mean of the distribution due to the stimulation. Here $\tilde{\mu}_{post}$ is a constant quantity; thus, the distribution will not change after settling on the stable distribution due to the stimulation. This would be a reasonable assumption for the timescales observed in the 15 spine stimulation (Fig. 3). However, longer timescales or a different number of stimulation events may not exhibit this stable behavior. Instead, we see a decay back to the baseline for the seven-spine experiment (see Supplementary Fig. 6). In the model, we can account for this decay back to the original distribution by introducing a time-dependent $\tilde{\mu}_{post}$ as follows

$$\tilde{\mu}_{post}(t) = \tilde{\mu}_{pre} + \Delta\mu e^{-\frac{t}{\tau}} \quad (16)$$

where $\tau$ can be considered to be the relaxation time back to the pre-stimulation baseline after a stimulation event. We hypothesize that $\tau$ is related to the number of stimulations and is much larger than the timescales we considered in this experiment.

**Reporting summary**. Further information on research design is available in the Nature Portfolio Reporting Summary linked to this article.

## Data availability

Experimental data sets included in the manuscript to generate the figures can be found in the following public github repository https://github.com/meggl23/SpontaneousSpines with https://doi.org/10.5281/zenodo.8183975. A part of the original data in this paper has previously been analyzed in a separate preprint to derive a model for multi-spine stimulation[59].

## Code availability

Experimental code to generate the figures in this plot can be found at the github repository https://github.com/meggl23/SpontaneousSpines with https://doi.org/10.5281/zenodo.8183975[60]. The code used to generate the spine metrics from the images can be found at https://github.com/meggl23/MultiSpinePlasticity with https://doi.org/10.5281/zenodo.7691901[57].

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

## Acknowledgements
This research was supported by University of Bonn Medical Center, University of Mainz Medical Center, ReALity program at the Mainz Medical Center, the German Research Foundation via CRC1080 (to J.P., T.T.), RIKEN Center for Brain Science, JSPS Core-to-Core Program (JPJSCCA20220007 to Y.G.). This project has received funding from the European Research Council (ERC) under the European Union's Horizon 2020 research and innovation program ("MolDynForSyn," grant agreement No. 945700). This research was also supported by an add-on fellowship of the Joachim Herz Stiftung (to M.F.E.). T.T. and Y.G. thank all our group members for fruitful discussions and Pietro Verzelli and Carlos Wert Carvajal for feedback on an earlier version of the manuscript (T.T.).

## Author contributions
M.F.E. and J.P. analyzed the dataset and developed the model; T.C. conducted the experiments; M.F.E., J.P., T.C., Y.G. and T.T. prepared the manuscript; M.F.E. and T.T. conceived study. Y.G. and T.T. supervised the project.

## Funding

## Competing interests
The authors declare no competing interests.
