## [Peer Review File · Communications Biology]

Reviewers' comments:

Reviewer #1 (Remarks to the Author):

Major Comments:

1) The most important comment is, the links of the model to plausible biological mechanisms need to be emphasized and clarified. First – minor but important - On page 5 below Eq. 2, the sentence "(See the supplemental...biological mechanisms) should have the parentheses removed, and be written more strongly as "For plausible links to biological mechanisms, see the Supplementary Material". Next, down in the supplementary, page 19 – biological mechanisms paragraph, line 1 – please add, next to the Shomar et al citation, a citation to Bonilla-Quintana et al 2020. Because, on page 20, this paper is needed, as follows. The suggestion on page 20 that molecules that "bind/unbind the spine wall" could "reverse" their action in a way that makes spine size oscillate is completely implausible from thermodynamics/statistical mechanics. This is not how binding processes generally progress! Such processes go instead to a steady state, unless they are driven out of equilibrium by an energy-consuming process which, here, would itself have to be oscillating. I read over some of your references. It looks like the model of Bonilla-Quintana et al. 2020 provides a much more plausible explanation for spine size oscillations! In that paper, an oscillatory component of spine size, with periods on the order of 10 minutes or so, similar to your data, seems clearly present in several figures. Please rewrite the "binding" paragraph on page 20 to, most likely, discuss that Bonilla model.

2) Discussion, second paragraph. The "degrades previously learned patterns..." phrase needs either a caveat, or elimination, because the data in this study only cover ~1 hour and do not address how memories might be preserved, or eliminated, over longer times. Either eliminate this phrase or, if you want to keep it, add a caveat. A possible example (one of many possibilities), "Our data do not, however, address how some memories, and possibly a sparse subset of large spines, can be preserved over timescales of days or longer".

3) Page 2 last line, rewrite as "The spine size changes are robustly negatively correlated between neighboring time steps ". It would also greatly help the reader if a sentence was added connecting this correlation to the "oscillatory dynamics" on page 4, because currently no justification of "oscillatory" is given. I suggest adding "This negative correlation suggests an oscillatory dynamic component".

4) The paragraph discussing Figs. 3C and 3D, bottom of page 5 and top of page 8, is incorrect and needs rewriting. For Fig. 3C (homosynaptic spines only) the top portion shows that the stimulation does in fact shift the location of the Gaussian plot (teal) although it does not change its shape. There is no "inset" in Fig. 3C. There is a bottom portion that shows a single dynamic plot (not plots). For Fig. 3D, the description should read similar except that here, neither the location nor shape of the gaussian plot shift upon stimulus and the dynamic plot does not show a stimulus effect.

5) Figure 2 legend, the panel i description actually refers to panel j, and the panel j description to panel i.

6) Last sentence of the Results section (page 10). Add the caveat ", at least over the time scales considered in this study". This caveat is necessary because "any model" is a very strong statement and because the limitation of an hour-long time scale for this study is not otherwise mentioned in the Discussion. Alternatively, this caveat could be addressed in the Discussion.

Minor Comments:

- 1) Please go through manuscript carefully again. As I detail in the following comments, I have found numerous minor English errors, the panels in one legend are switched from their correct order, and a couple equation numbers are incorrect, etc. It is likely that there are other minor errors I did not find.
- 2) The legend to Fig. 1b says this spine is exhibiting both growth and shrinkage in the time frame

shown. But, unless the panels in Fig. 1b are not time ordered, only shrinkage is shown.

3) The description of Figure 1c appears incorrect and the figure and description should be revised. The legend states that the blue and red curves are each for different timepoints, but it looks to me as if the red curve is a mean/average of the curves for the individual timepoints. Also more than two colors should be used here. Each curve should have a distinct color.

4) Page 2 first line, remove "given"

5) The supplemental figures are currently not ordered correctly. Figure S4 is the first figure referenced in the main text, it should therefore be changed to Figure S1, and the other figures (and all citations of them) relabeled according to when they are first referenced in the text. This should be done after any other revisions to supplemental figures according to reviewer's comments. Note that in my other comments, I am using the current (incorrect) supplemental figure labels.

6) Add a space between number and measurement unit throughout, that is, write "0 mM" instead of "0mM"

7) Page 2 six lines from bottom, rewrite as "Notably, we observed the mean size of the spine population is also remarkably stable..."

8) Page 2 three lines from bottom, the reader cannot really tell which time points the +2 and +10 refer to. List the numbers for the time points explicitly (e.g. 2, 10, 18 ... min... It seems to me the list should end at 55, or there should be a difference of 55 between the first and the last...) , either here, or else in the Figure 1 legend. If the latter, refer the reader to the legend here.

9) Figure 1 legend last line, the differences are significant yes? So remove "no" from in front of "significant".

10) Page 4 first line, should be "approximately 10 min" instead of "10s of min" because the correlations are between neighboring time points

11) Page 4, replace "skewed, log-normal-like profile" with "is a skewed, log-normal like profile" and put parentheses around the figure label.

12) The very next line, replace "the new distributions" with "the distributions of changes". This is necessary because to the reader, the "new distributions" means the new size distributions, which is not what you mean.

13) Towards bottom of page 4, rewrite as "Equation (7) in Methods" (or whatever is the standard style for this journal, e.g. Eq. vs. Equation)

14) Near top of page 5, point 2. It is very confusing to the reader to say an OU term was "reintroduced", because in the Methods, the negative momentum term, which is already present, is described as a type of OU process. It would be better to state "Therefore, we also add an OU global drift term".

15) Page 8, 11 lines down, a wrong figure panel is cited. Replace "Fig. 3e and Fig. 3h" with "Fig. 3f and Fig. 3h".

16) Page 8, five lines from bottom. The sentence "Given the..." is written incorrectly. It should read "Furthermore, given the observation that small spines are most affected by stimulation, we examined the effect of only changing the parameters of the smallest spines in the model (Fig. 4h)."

17) Further down on page 8, replace "for the case where only seven spines were stimulated" with "for a separate case in which only seven spines were stimulated". (this is correct English because this is the first time you mention this case).

18) Page 11 near top, "large spines are the reservoirs" is too strong, "large spines are likely to be the reservoirs" is correct.

19) Discussion, second paragraph, change "may preventing" to "may prevent".

20) Make sure both Ornstein and Uhlenbeck are capitalized wherever they appear

21) In Methods, Numerical, first sentence after Image Analysis – Replace "cite" with a real citation and add to the bibliography

22) On page 13, "Fig. S1e-f" is cited, but there is no panel f in Fig. S1!

23) Legend to Figure S2, correct Equation (1) to Equation (7).

24) Page 19, biological mechanisms paragraph, line 3 – I suggest reminding the reader that size in this study is quantified by spine area, not volume, by writing something like "...Consider a spine of a given size (quantified in this study by area, see Methods)..."

25) Figure S3 legend, line 4, replace "with 4um" by "within 4 um".

Reviewer #2 (Remarks to the Author):

Eggl et al present a experimental and computational model-based study quantifying dendritic spine fluctuations in in vitro organotypic hippocampal slices. This study complements and extends related prior work with several novel and interesting aspects: 1) the focal stimulation and induction of plasticity in a subset of spines; 2) the ~ 1 hour timescale of measurements contrasts with slower timescales of other studies; 3) the systematic development of a simple abstract model to account for the data; 4) the finding that most of the shift in spine size distributions can be accounted for by changes in small spines. This will be a valuable addition to the literature. However I also have some concerns about the manuscript described below.

Major comments

- Can the authors estimate what percentage of fluctuations may be due to observation noise? E.g. due to the limited spatial resolution of two-photon microscopy, or spines moving in and out of the focal plane, etc. This may be contributing to their activity independent noise estimates.
- Can the authors elaborate on the benefit of calculating the entropy of these distributions? What does it tell us about brain learning? Particularly since the entropy of a gaussian distribution (which these distributions approximate) is just a simple function of its standard deviation. What does the entropy measure tell us beyond: "the distributions get broader following LTP stimulation"?
- The non-markovian/ocillatory dynamics is interesting. I wonder if the authors could speculate on possible biological mechanisms for this, and also what the functional implications may be. Ideas in this direction could motivate further studies.

Minor comments

- pg 4. In the numerical list of key features in the data, does 1 not automatically imply 2? i.e. if the distribution is stable then the mean must also be stable.
- pg 4. "However, we still do not use the experimental data's optimal fits". I find it difficult to understand this sentence, perhaps it could be reworded for clarity?
- pg 4. I found the paragraph describing various model extensions very hard to follow. I wonder if it could be restructured for clarity, perhaps by giving the various model versions different labels or numbers.
- Figure 3h. It is interesting that the distribution of fluctuations is narrower for small unstimulated spines compared to baseline fluctuations. Does this imply some type of suppression of activity-independent fluctuations in these spines?
- Discussion "we saw that small spines act as the means of acquiring new memories during induction". This seems like an overly strong conclusion, since some of the large spines do change, including reducing in size. And to my mind, decreases in synaptic strength are just as valid a mechanism for storing information as increases.
- Discussion pg 10, the authors say "this activity-independent plasticity is an energy-efficient (homeostatic) mechanism". It is not clear to me why it would be energy-efficient. If anything I would assume constant remodelling of large spines is energetically more expensive than it would be to remodel small spines if we assume energetic costs scale with the amount of actin or other proteins

involved. Can the authors justify this idea more?

- Methods pg 12, "the strength of that synapse cite". Presume there is a missing reference to be inserted instead of the word "cite"?

- Methods p13, says "The OU process was used by Loewenstein et al. (2011) to model activity-independent plasticity in a framework with multiplicative, instead of additive, noise, and takes the following form". However the noise term in eq(6) is additive, not multiplicative.

- Methods p14, (also eq2 in main text) I find the notation used for $\mu_{\log,i}$ and $\sigma_{\log,i}$ a bit misleading since i is indexing time, I initially thought that this implied that these parameters are also time-varying, however it seems they are actually just functions of V (which of course changes with time).

- Methods pg 14, I may have misunderstood but there appears to be an inconsistency between eqs 11+12 vs 13+14. The $\mu_s(V)$ in eq 11 has a delta term on the r.h.s., but in eq 13 both μ and δ are present in the numerator on the r.h.s. Is the delta term absorbed into μ_s or not?

- Methods pg 15, the text in the "Details of the data set" section implies it will link to the source for the data and code but the link is not provided there at all, but in a later section "Data and code availability and history".

- Figure S1a caption says "black boxes", but these are not observable in the image.

- It might be worth also citing the following two papers who found that large spines were more stable than small spines via longitudinal in vivo 2-photon imaging:

Grutzendler, J., Kasthuri, N. and Gan, W.B., 2002. Long-term dendritic spine stability in the adult cortex. *Nature*, 420(6917), pp.812-816.

Holtmaat, A.J., Trachtenberg, J.T., Wilbrecht, L., Shepherd, G.M., Zhang, X., Knott, G.W. and Svoboda, K., 2005. Transient and persistent dendritic spines in the neocortex in vivo. *Neuron*, 45(2), pp.279-291.

- And potentially also of interest are the following theoretical papers which propose biophysical mechanisms by which larger spines/synapses could be more stable than small spines/synapses:

Shouval, H.Z., 2005. Clusters of interacting receptors can stabilize synaptic efficacies. *Proceedings of the National Academy of Sciences*, 102(40), pp.14440-14445.

O'Donnell, C., Nolan, M.F. and van Rossum, M.C., 2011. Dendritic spine dynamics regulate the long-term stability of synaptic plasticity. *Journal of Neuroscience*, 31(45), pp.16142-16156.

Bell, M.K., Holst, M.V., Lee, C.T. and Rangamani, P., 2022. Dendritic spine morphology regulates calcium-dependent synaptic weight change. *Journal of General Physiology*, 154(8), p.e202112980.

Jozsa, M., Donchev, T.I., Sepulchre, R. and O'Leary, T., 2022. Autoregulation of switching behavior by cellular compartment size. *Proceedings of the National Academy of Sciences*, 119(14), p.e2116054119.

Typos

- page 2, parentheses oddly placed: "(e.g.) lateral diffusion, active trafficking, endocytosis, and

exocytosis(Choquet and Triller, 2013; Ziv and Fisher-Lavie, 2014))"

- There is something ungrammatical about this sentence: "However, given the fact that synaptic changes are driven by molecular processes that are inherently noisy (e.g.) lateral diffusion, active trafficking, endocytosis, and exocytosis, implies that such spontaneous changes are inevitable."

- pg 4 "a positive mean change and a have smaller standard deviation", extra "a".

- pg 7 "are not significantly from", missing "different"

- pg 8 " during for the homo-", extra "for"

- pg 8 "that the spines became are more", extra "are"

- pg 10, "Our model relies a fast and slow mechanism", missing an "on"?

- pg 10, "fast oscillatory changes may preventing", -> prevent?

- pg 10, "and thus have no little to no", extra "no".

- pg 12, "in the case were these", were -> where

- pg 14, "where we use set the shift", extra "use"

- pg 12, "Briefly, he spine ROI", he -> the

Reviewer #3 (Remarks to the Author):

In this study the authors imaged dendritic spines of GFP-expressing CA1 pyramidal neurons in hippocampal organotypic slices in the presence of TTX, picrotoxin and no added Mg²⁺. Images were obtained at 8 time points for overall durations of 55 minutes. In the course of these experiments, a subset of spines was stimulated (potentiated) by 2-photon glutamate uncaging or sham-uncaging (that is in the presence of absence of MNI-glutamate) after the second time point. Volumes of individual spines were then estimated at each of the time points. The authors then step through a series of increasingly elaborate stochastic models aimed to capture the dynamics of size changes, size distributions and the effects of stimulation on these dynamics and distributions, in order to obtain a new understanding of how both spontaneous synaptic fluctuations and local plasticity contribute to population-level synaptic dynamics.

There are many things to like about this study, in particular, the focus on short term dynamics. That is, while there have been quite a few attempts to model the activity independent size dynamics of dendritic spines over many hours and days, few studies have focused on the short-term dynamics (minutes). Moreover, the incorporation of rapid potentiation events into such models has not been attempted as far as I know.

Having said this, there are numerous matters that to my mind need to be addressed.

- 1) It would be much easier to follow the text if the explicit models used for Fig. 2 were placed in the Results rather than in the Methods section (Numerical), as essentially, the rationale for the development of the models and their evaluation appears twice – once in the main text and once in the Methods. Consequently, one has to jump back and forth between the two sections, which make the logic rather difficult to follow, in particular given the use of multiple terms for the same components (see below). I would suggest to merge everything into the Results, into a single coherent exposition.
- 2) I found the model to be fairly complicated in comparison to prior models. In the models put forward in the past (e.g. the Kesten process (Statman et al., 2014), or a non-linear Langevin process (Yasumatsu et al., 2008; Hazan and Ziv, 2020), log-normality (or something close to it) emerged from

a set of simple assumptions backed by data. Here, unless I am missing something, single time point changes are assumed a-priori to have log-normal distributions. Consequently, the model “plugs in” log normality through assuming that size changes are sampled from a log-normal distribution calculated separately (or approximately so) for each synapse, based on its momentary size. I am not sure I follow the rationale for this elaborate modeling process (which might relate to point 1 above).

3) Figure 1b and d. These fits are based on prior studies, among which is Hazan and Ziv 2020. In that study, a linear relation was (predicted and) observed between the *variance* and synaptic size *squared*. This is important if the function has a constant term, which the less-than-perfect fit to a linear function in Fig 1b suggests it does. One wonders if the fit would be better if the values were extracted from the squared function, allowing to do away with the need for “altering the “best” linear fits of the means and standard deviations”

4) This model adds a component which was not incorporated into prior models, the “negative momentum”. If I understand correctly, this term effectively implies that each synapse has some unique “set point”, or “short term memory” that acts to counter random changes on short time scales (minutes). Is this so? It is interesting that on longer time scales, no evidence was found for this component (Fig. 1F; see also Minerbi et al., 2009, Fig. S7). Could the authors add a plot similar to Fig. 1g for longer time intervals?

5) Equation (2) is the model ultimately derived in this manuscript. It would be useful to stick to the same terminology for each of its components as this would profoundly facilitate comprehension. For example, the first component is referred to as “Long-term stochasticity” but also as “linear fits”, “stochastic log-normal component”, the second is referred to as “drift” but also as “amendments to the fits” (and in Fig 4f “Short-time drift change”), and the third term is referred to as “Correlation” but also as “negative momentum” and “mean-reverting” (unless, admittedly, I failed to understand – please see point #1).

6) In the attempt to alter the model parameters to fit the stimulation time point, an assumption is made that “the correlation term is not active during the stimulation”. What is the justification? This feels rather arbitrary. In addition, in the supplementary discussion it is stated that “As we did not need to alter the correlation component, we assume, intuitively, that this component is unaffected by the stimulation” so this is a bit confusing: Was it ultimately active or inactive?

7) “For the homosynaptic spines in Fig. 4d a different behavior emerges. We see that the standard deviation is tilted upwards, meaning that the resulting log-normal distribution has increased its standard deviation and that the spines become more variable during stimulation. This increase in the standard deviation is only observed in the medium-sized bins and not for the small or large spines”. A) Given that synapses are rapidly enlarged by the potentiation protocol, isn’t the increase in the standard deviation a trivial/expected outcome? B) Given that the Methods section states that “Medium-sized spines with a clear spine head within the field of view were preferentially targeted for stimulation”, wasn’t the selective effect on medium sized synapses expected?

8) Discussion – Indeed, prior studies assumed ergodicity, but the model proposed in the current study is not free of it. For example, the “fits to mean changes” (e.g. Fig 2b) imply that synaptic sizes converge toward some mean size, irrespective of their specific identity. Also, the changes they experience (the Long-term stochasticity component) are obtained from distributions that only depend on momentary synaptic sizes. Similarly, the drift term in equation 2 implies something similar. If I understand correctly, the departure from ergodicity is mainly to be found in the Correlation component. This is interesting and perhaps worthy of discussion. It is also worth mentioning that given the short observation periods, the lack of evidence for ergodic behavior is not completely unexpected.

Minor comments –

1) The term “turnover” should probably be avoided as neither synapse nor synaptic molecule turnover was studied here.

2) Some justification for the use of the terms homosynaptic and heterosynaptic should probably be provided.

3) The methods mention that “For a subset of experiments, calcineurin or CaMKII was inhibited with FK506 (2 μ M, Tocris) or myristoylated Autocamtide-2-related inhibitory peptide (AIP, 5 μ M, Calbiochem) respectively”. Is this information correct? Did these agents affect the results?

4) Supplementary discussion – it is suggested that the correlation term acts as fast time-scale mechanism which is only overcome with activity-dependent changes that allow the spines to grow into the sizes that fall on the long tail of the distribution. I note here that heavy tail distributions of spine sizes emerge in networks that never experienced any activity whatsoever (see for example Hazan and Ziv 2020), so some care should be made in this regard.

Reviewer identifies himself as Noam Ziv

Dear Referees, dear Dr. Chong,

We sincerely thank the referees for their invaluable feedback and insightful comments on our manuscript. The constructive criticisms and suggestions have helped improve the revised manuscript. In this response, we provide a point-by-point reply (**in bold**).

Reviewer #1 (Remarks to the Author):

Major Comments:

1) The most important comment is, the links of the model to plausible biological mechanisms need to be emphasized and clarified. First – minor but important - On page 5 below Eq. 2, the sentence “(See the supplemental...biological mechanisms) should have the parentheses removed, and be written more strongly as “For plausible links to biological mechanisms, see the Supplementary Material”.

We thank the reviewer for this helpful point, which we have corrected as suggested. The highlighted sentences (on lines 161-162 in the revised manuscript) now read: “For plausible links to biological mechanisms, see the Supplementary Material”.

Next, down in the supplementary, page 19 – biological mechanisms paragraph, line 1 – please add, next to the Shomar et al citation, a citation to Bonilla-Quintana et al 2020. Because, on page 20, this paper is needed, as follows. The suggestion on page 20 that molecules that “bind/unbind the spine wall” could “reverse” their action in a way that makes spine size oscillate is completely implausible from thermodynamics/statistical mechanics. This is not how binding processes generally progress! Such processes go instead to a steady state, unless they are driven out of equilibrium by an energy-consuming process which, here, would itself have to be oscillating. I read over some of your references. It looks like the model of Bonilla-Quintana et al. 2020 provides a much more plausible explanation for spine size oscillations! In that paper, an oscillatory component of spine size, with periods on the order of 10 minutes or so, similar to your data, seems clearly present in several figures. Please rewrite the “binding” paragraph on page 20 to, most likely, discuss that Bonilla model.

We thank the reviewer for the insightful comment and valuable reference. As suggested, we have included references to the Bonilla-Quintana model and additional articles describing/modeling wave-like dynamics of the actin cytoskeleton network, which can be found on lines 607-613 in the revised manuscript. Additionally, we have expanded our discussion of the possible biological mechanisms of the negative correlations we measured. Given these studies, the negative-momentum term in our model implicitly takes into account the minute-long oscillations that result from the interactions between actin rigidity, spine membrane surface tension, and stochastic actin nucleation.

2) Discussion, second paragraph. The “degrades previously learned patterns...” phrase needs either a caveat, or elimination, because the data in this study only cover ~1 hour and do not

address how memories might be preserved, or eliminated, over longer times. Either eliminate this phrase or, if you want to keep it, add a caveat. A possible example (one of many possibilities), “Our data do not, however, address how some memories, and possibly a sparse subset of large spines, can be preserved over timescales of days or longer”.

Thank you for the helpful suggestion. We now include an additional sentence: “We note that our results do not explain how a small subset of spines (e.g., large spines representing selected memories) can be preserved over timescales of days or months (for a brief discussion on how longer timescales could be incorporated in the model, see the numerical methods section).” (lines 293-295 and method section starting at line 446)

3) Page 2 last line, rewrite as “The spine size changes are robustly negatively correlated between neighboring time steps “. It would also greatly help the reader if a sentence was added connecting this correlation to the “oscillatory dynamics” on page 4, because currently no justification of “oscillatory” is given. I suggest adding “This negative correlation suggests an oscillatory dynamic component”.

We agree and have therefore added the comment “This negative correlation suggests an oscillatory dynamic component” to the corresponding sentence. (line 115)

4) The paragraph discussing Figs. 3C and 3D, bottom of page 5 and top of page 8, is incorrect and needs rewriting. For Fig. 3C (homosynaptic spines only) the top portion shows that the stimulation does in fact shift the location of the Gaussian plot (teal) although it does not change its shape. There is no “inset” in Fig. 3C. There is a bottom portion that shows a single dynamic plot (not plots). For Fig. 3D, the description should read similar except that here, neither the location nor shape of the gaussian plot shift upon stimulus and the dynamic plot does not show a stimulus effect.

We thank the reviewer for noticing the error. We had mistakenly omitted the references to Figs. 3C and 3D. We have corrected this and rewritten the section to highlight the change in the homosynaptic spines. We have also removed the reference to an “inset” and instead referred to it as the “figure below”. (lines 181-185)

5) Figure 2 legend, the panel i description actually refers to panel j, and the panel j description to panel i.

The reviewer is completely correct; the descriptions have been revised such that they now refer to the correct figure.

6) Last sentence of the Results section (page 10). Add the caveat “, at least over the time scales considered in this study“. This caveat is necessary because “any model” is a very strong statement and because the limitation of an hour-long time scale for this study is not otherwise mentioned in the Discussion. Alternatively, this caveat could be addressed in the Discussion.

We acknowledge that our previous statement was too strong and have added the caveat that our study is limited to shorter time scales as follows: “..., at least over the time scales considered in this study.” (lines 265)

Minor Comments:

1) Please go through manuscript carefully again. As I detail in the following comments, I have found numerous minor English errors, the panels in one legend are switched from their correct order, and a couple equation numbers are incorrect, etc. It is likely that there are other minor errors I did not find.

We thank the reviewer for the careful reading of our manuscript. We have gone through multiple rounds of proofreading of the revised manuscript to improve it.

2) The legend to Fig. 1b says this spine is exhibiting both growth and shrinkage in the time frame shown. But, unless the panels in Fig. 1b are not time ordered, only shrinkage is shown.

The reviewer is correct in that the previous arrangement by the size of the spine, inadvertently only implied shrinkage. We have now ordered this panel in time, which shows growth and shrinkage (see below):

Fig 1b: Example of spontaneous dynamics at the single spine level. The spine (marked by a gray rectangle in a) exhibits both growth and shrinkage in the observed time frame.

3) The description of Figure 1c appears incorrect and the figure and description should be revised. The legend states that the blue and red curves are each for different timepoints, but it looks to me as if the red curve is a mean/average of the curves for the individual timepoints. Also more than two colors should be used here. Each curve should have a distinct color.

We agree with the reviewer that Fig. 1c and its description in the submitted manuscript did not clearly demonstrate what we aimed to show. Our original goal was to illustrate the stability of the size distribution by plotting the distributions of each of the timesteps (with a color scheme ranging from blue to red, as in Fig. 1d) on the same figure. We have now corrected this by plotting each distribution as gray lines, changing the inset lines to

be dashed, and updating the corresponding legend. We have also changed the figure description. We hope this change alleviates the reviewer's concern and clarifies the message. See the revised Fig. 1c below:

Fig 1c: The spine sizes follow a temporally stable right-skewed distribution with a long tail. Each gray line refers to a different snapshot distribution, which shows significant overlap. Inset: The mean size of the full spine population (red) is shown across time along with the dynamics of selected spines (gray) at each time point, where the time points are at (-15, -10, -5, 2, 10, 20, 30 and 40 minutes).

4) Page 2 first line, remove “given”

We have removed the word “given”.

5) The supplemental figures are currently not ordered correctly. Figure S4 is the first figure referenced in the main text, it should therefore be changed to Figure S1, and the other figures (and all citations of them) relabeled according to when they are first referenced in the text. This should be done after any other revisions to supplemental figures according to reviewer's comments. Note that in my other comments, I am using the current (incorrect) supplemental figure labels.

We have reordered supplemental figures as suggested: old Figure S4 is now S1, S2 is a new figure, old S2 is now S3, old S1 is now S4, S5 is a new figure suggested by another reviewer, and old S3 is now S6.

6) Add a space between number and measurement unit throughout, that is, write “0 mM” instead of “0mM”

We have revised the notation throughout the manuscript accordingly.

7) Page 2 six lines from bottom, rewrite as “Notably, we observed the mean size of the spine population is also remarkably stable...”

We have altered the sentence by removing the word “exhibits”.

8) Page 2 three lines from bottom, the reader cannot really tell which time points the +2 and +10 refer to. List the numbers for the time points explicitly (e.g. 2, 10, 18 ... min... It seems to me the list should end at 55, or there should be a difference of 55 between the first and the last...), either here, or else in the Figure 1 legend. If the latter, refer the reader to the legend here.

Thank you for spotting the potential confusion about reference to the time point when the distribution statistically differs (which is the change in difference from time point +10 to +20). We implemented the following changes to address the issue: i) added a star to the corresponding distribution as well as axis labels of Fig. 1d, ii) included a sentence on this in the figure description, iii) added a detailed description of the time-points in the paragraph above (lines 73 and 74) and iv) have incorporated a supplemental figure, which shows an alternative depiction of the spine size distributions of 1f and highlights the differences of the one statistically significant distribution. We have also amended the reference to 55 min, with the aim to clarify it. More precisely, the experiments were performed over 55 min, including a period of 15 min before the laser exposure. The new Fig. 1d and Fig. S2 is shown below:

Fig. 1d: Collective distributions of the spine size changes (Δs) from time point to time point follow a Gaussian distribution. The black lines denote the corresponding Gaussian fits. The * denotes the single distribution that is significantly different.

Fig. S2: Alternative depiction of the spine changes distributions (which display Gaussian features) from Fig. 1d. The distribution of spine size changes of activity-independent plasticity do not differ from each other significantly (K.S. test), apart from the single change distribution from time point +10 to +20, which is marked with a red line.

9) Figure 1 legend last line, the differences are significant yes? So remove “no” from in front of “significant”.

In this case, we wish to show that the black line is statistically not different from the dataset, we thus cannot rule out the possibility that the data is generated by the lognormal distribution. However, we agree with the reviewer that the phrasing is not completely clear, so we have instead changed the phrase to “no statistical difference seen between the dataset and a lognormal distribution”. We hope this alleviates the reviewer’s concern.

10) Page 4 first line, should be “approximately 10 min” instead of “10s of min” because the correlations are between neighboring time points

We concur with the reviewer’s suggestion to change the phrase “10s of min”. (line 111)

11) Page 4, replace “skewed, log-normal-like profile” with “is a skewed, log-normal like profile” and put parentheses around the figure label.

We thank the reviewer for this point. The text has been changed as suggested (line 121).

12) The very next line, replace “the new distributions” with “the distributions of changes”. This is necessary because to the reader, the “new distributions” means the new size distributions, which is not what you mean.

We fully agree and have clarified our intention by changing “new distributions” to read “distribution of changes”. (line 122)

13) Towards bottom of page 4, rewrite as “Equation (7) in Methods” (or whatever is the standard style for this journal, e.g. Eq. vs. Equation)

We have added “Equation” to the sentence highlighted by the reviewer. (line 140)

14) Near top of page 5, point 2. It is very confusing to the reader to say an OU term was “reintroduced”, because in the Methods, the negative momentum term, which is already present, is described as a type of OU process. It would be better to state “Therefore, we also add an OU global drift term”.

We thank the reviewer for the excellent suggestion. We have corrected the statement, which now reads: “Therefore, we also add a global OU drift term (referred to as Drift below)”. (line 151)

15) Page 8, 11 lines down, a wrong figure panel is cited. Replace “Fig. 3e and Fig. 3h” with “Fig. 3f and Fig. 3h”.

We thank the reviewer for spotting this, we have now corrected it in the revised manuscript.

16) Page 8, five lines from bottom. The sentence “Given the...” is written incorrectly. It should read “Furthermore, given the observation that small spines are most affected by stimulation, we examined the effect of only changing the parameters of the smallest spines in the model (Fig. 4h).”

We are grateful for the reviewer’s valuable suggestion and have changed the corresponding sentence to: “Furthermore, given the observation that small spines are most affected by stimulation, we examined the effect of only changing the parameters of the smallest spines in the model (Fig. 4h).” (line 250-251)

17) Further down on page 8, replace “for the case where only seven spines were stimulated” with “for a separate case in which only seven spines were stimulated”. (this is correct English because this is the first time you mention this case).

We have changed the phrase “for the case where” to “for a separate case in which”. (line 238)

18) Page 11 near top, “large spines are the reservoirs” is too strong, “large spines are likely to be the reservoirs” is correct.

We agree with the reviewer and have toned down the statement as suggested. (line 312)

19) Discussion, second paragraph, change “may preventing” to “may prevent”.

Corrected.

20) Make sure both Ornstein and Uhlenbeck are capitalized wherever they appear.

We apologize for this oversight and have capitalized the names throughout the text.

21) In Methods, Numerical, first sentence after Image Analysis – Replace “cite” with a real citation and add to the bibliography

We have added Chen et al., 2013 and Bartol Jr et al., 2015 citations to this sentence and the bibliography.

22) On page 13, “Fig. S1e-f” is cited, but there is no panel f in Fig. S1!

We have removed the reference to figure S1f .

23) Legend to Figure S2, correct Equation (1) to Equation (7).

The reference to equation (1) in Fig. S2 (which is S3 in the revised manuscript) has been replaced with a reference to equation (9), which is equation (7) in the original manuscript.

24) Page 19, biological mechanisms paragraph, line 3 – I suggest reminding the reader that size in this study is quantified by spine area, not volume, by writing something like “...Consider a spine of a given size (quantified in this study by area, see Methods)...”

We concur with the point raised by the reviewer, and we have modified the relevant sentence in the paragraph as suggested. (line 593)

25) Figure S3 legend, line 4, replace “with 4um” by “within 4 um”.

We have altered the legend of figure S3 (S6 in the revised manuscript) as suggested by the reviewer.

In summary, we thank reviewer 1 for the insightful and valuable comments to improve the strength and accessibility of our manuscript.

Reviewer #2 (Remarks to the Author):

Eggl et al present a experimental and computational model-based study quantifying dendritic spine fluctuations in in vitro organotypic hippocampal slices. This study complements and extends related prior work with several novel and interesting aspects: 1) the focal stimulation and induction of plasticity in a subset of spines; 2) the ~1 hour timescale of measurements contrasts with slower timescales of other studies; 3) the systematic development of a simple abstract model to account for the data; 4) the finding that most of the shift in spine size distributions can be accounted for by changes in small spines. This will be a valuable addition to the literature. However I also have some concerns about the manuscript described below.

Major comments

- Can the authors estimate what percentage of fluctuations may be due to observation noise? E.g. due to the limited spatial resolution of two-photon microscopy, or spines moving in and out of the focal plane, etc. This may be contributing to their activity independent noise estimates.

Thank you to the reviewer for this helpful comment. When acquiring the data, care was taken to ensure that the entire dendrite (along with its spines) was included in every z stack at every time. This was achieved by capturing blank z steps above and below the dendrite. Spine dynamics do cause movement between images, and sometimes a spine will move “behind” another spine (or the dendrite) from the perspective of the microscope detector. Any spine that could not be clearly identified at every time point was not included in the analysis. Moreover, at a z step size of $0.5 \mu\text{m}$, most spines appear in 4-5 z steps, allowing us to capture the entire spine volume.

To estimate noise within the microscope system, we biolistically transfected neurons in organotypic slices with GFP, then fixed the slices and imaged them as usual. Fixed spines imaged over 1 hour showed minimal differences between frames (please see the example below), suggesting that the contribution of noise is extremely small. Measuring spine intensity from frame to frame, we found a difference in brightness of 0.61% (s.e.m., 0.07), indicating that the system itself is contributing less than 1% of total noise.

Example of fixed spine imaging. Numbers in the bottom left of images indicate the time in minutes during the experiment. Scale = $1 \mu\text{m}$

- Can the authors elaborate on the benefit of calculating the entropy of these distributions? What does it tell us about brain learning? Particularly since the entropy of a gaussian distribution (which these distributions approximate) is just a simple function of its standard deviation. What does the entropy measure tell us beyond: "the distributions get broader following LTP stimulation"?

We thank the reviewer for this interesting question, and we agree that entropy is a simple function of the standard deviation in the case of the Gaussian distribution. However, the entropy measure also goes beyond simply indicating the broadening of distributions following LTP stimulation, which is why it continues to attract significant interest from the neuroscientific community in the context of circuit computation (Strong et al., 1998), synaptic memory (Samavat et al., 2022), epilepsy detection (Cao et al., 2004) as well as being a source for model design (e.g., maximum entropy models, Granot et al., 2013). Entropy being a mathematical measure of the information-carrying capabilities of a network, describes the amount of disorder or uncertainty about possible states a system can assume. Following LTP stimulation, we observed an increase in the range of synaptic sizes, thus, a larger set of possible states consistent with higher entropy. This higher entropy could facilitate learning by enabling the network to differentially encode a wider range of inputs. Secondly, entropy can also reflect the stability and robustness of synaptic connections. A higher entropy, reflected by a more diverse distribution of synaptic strengths, could make a network less sensitive to changes in individual synapses. This increased ability to buffer against noise or disruptions, such as the loss or weakening of specific synapses, helps promote the overall robustness of the network.

The robust increase in spine sizes, which is reflected in the change in entropy post stimulation shows that LTP induction has provided the network with higher information-carrying capabilities. We have now incorporated this discussion of entropy into the revised discussion section.

We expanded the discussion section (lines 321-329) to accommodate the above points.

- The non-markovian/ocillatory dynamics is interesting. I wonder if the authors could speculate on possible biological mechanisms for this, and also what the functional implications may be. Ideas in this direction could motivate further studies.

We are grateful to the reviewer for raising this interesting point. The non-Markovian (negative-momentum) term, introduced to recover the anti-correlation observed between time-adjacent size changes, can be linked to the oscillatory behavior observed in actin networks. Several reports (Bonilla-Quintana 2020 (study mentioned by Reviewer 1), Holmes 2012; Veksler, 2009) have shown that actin cytoskeletal dynamics can support and undergo wave-like oscillations in their structure in time with a period of 1-100 s. In spines in particular, this dynamic remodeling of actin could be associated with the

complex interaction between actin filament rigidity, spine membrane surface tension, and the formation of new actin nucleation points in the network.

These spontaneous spine oscillations potentially prevent a "winner-takes-all" effect which could lead to uncontrolled growth/shrinkage. A possible functional implication is that the trend to reverse previous changes serves to suppress progressive spine size changes that are locked in one direction. The stimulation induced forces allow for a collective shift of the spines to larger sizes. Additionally, it is plausible that these oscillations provide the spines with a "set point" within biologically plausible bounds by pushing the spine sizes from both edges toward the middle. Therefore, given that natural random protein movement within the spines will lead to spine changes, these oscillations could provide a mechanism to help preserve a stable synaptic size average and thereby contribute to network stability.

Interestingly, the specific experimental conditions that give rise to these negative spine correlations are yet to be fully understood. For example, Minerbi et al. (2009), did not report a negative correlation when studying long-term random fluctuations in dissociated cortical cultures at 30-minute intervals. On the other hand, Yasumatsu et al. 2008, reported a slight negative correlation (-0.189) when considering the change in hippocampal slice cultures over a period of days. From these and our results, two avenues of study arise. On the one hand, understanding the exact temporal relation between the oscillations and the size of the experimentally considered time bins would provide insight into whether the negative momentum and its amplitude does indeed preserve not only average synaptic size but helps maintain stable network and information retention properties. On the other hand, understanding the pharmacological conditions and cell environment controlling or inhibiting these oscillations (e.g. Ziv and Brenner, 2018; Hazan and Ziv, 2020) could provide insights into the mechanisms giving rise to or preventing negative correlations in synaptic size changes.

Once more, we are thankful for this interesting comment, and we have added the above discussion to the revised manuscript on lines 607-630.

Minor comments

- pg 4. In the numerical list of key features in the data, does 1 not automatically imply 2? i.e. if the distribution is stable then the mean must also be stable.

We have now removed feature 2 (the stable mean condition) and added it as a remark to feature 1 as follows: "As a consequence, the mean of the distribution needs to remain stable through time (Fig. 1c - inset)." (line 107)

- pg 4. "However, we still do not use the experimental data's optimal fits". I find it difficult to understand this sentence, perhaps it could be reworded for clarity?

We agree with the reviewer that the sentence could have been clearer. It now reads: “Despite the excellent agreement with the experimental results, we found it necessary to use the manually tuned fits for obtaining the mean and the standard deviation. As such, when implementing the Alt. fit LN model, we were not using the optimal fits shown in Fig. 2b.” (lines 142-144)

- pg 4. I found the paragraph describing various model extensions very hard to follow. I wonder if it could be restructured for clarity, perhaps by giving the various model versions different labels or numbers.

To this end, we have included additional phrases that provide names, which are “Best fit LN model”, “Alt. fit LN model”, and “LN-OU model” for the individual models in accordance with the names in Fig. 2. Additionally, certain sections from the numerical methods section were moved to the model derivation. We hope that these changes clarify this section.

- Figure 3h. It is interesting that the distribution of fluctuations is narrower for small unstimulated spines compared to baseline fluctuations. Does this imply some type of suppression of activity-independent fluctuations in these spines?

We thank the reviewer for raising this interesting point. We do indeed observe that the fluctuations are narrower for the small, unstimulated spines. This narrowing appears skewed to the right, such that the decrease in activity-independent fluctuations could be preferentially associated with the shrinkage of the small spines. In contrast to the stimulated small spines that undergo growth, neighboring small spines experience the stimulation only peripherally. In such a case, the components that induce growth may not reach levels sufficient to actually cause growth while they may be present at levels that could still counter (or compete with) activity-independent shrinkage.

We have added a comment reflecting the above points to the manuscript. (lines 202-208)

- Discussion "we saw that small spines act as the means of acquiring new memories during induction". This seems like an overly strong conclusion, since some of the large spines do change, including reducing in size. And to my mind, decreases in synaptic strength are just as valid a mechanism for storing information as increases.

We agree also with the reviewer that a decrease in synaptic strength is as valid a learning mechanism as growth in synaptic strength. However, in our experimental results (see Figs. 3f and 3h), we only observed a significant difference between baseline and stimulation in the smallest spines, specifically for their growth. We have changed the sentence on lines (284-287), it now reads:

"Additionally, in our experiments we observed that small spines preferentially showed a positive size change (Fig. 3f), and therefore they could act as points of information acquisition during plasticity induction. In contrast, large spines did not change their dynamics significantly after the stimulation, such that the large spines could help maintain the stability of previous state."

- Discussion pg 10, the authors say "this activity-independent plasticity is an energy-efficient (homeostatic) mechanism". It is not clear to me why it would be energy-efficient. If anything I would assume constant remodelling of large spines is energetically more expensive than it would be to remodel small spines if we assume energetic costs scale with the amount of actin or other proteins involved. Can the authors justify this idea more?

The reviewer makes an interesting point related to the energy efficiency of activity-independent plasticity. In our initial manuscript, we stated that this process is energy-efficient. However, as the reviewer correctly points out, constant remodeling of large spines, as demonstrated by the experimental data, would be energetically more expensive than remodeling small spines. More concretely, we hypothesize that the mechanism itself is not energy-efficient but that the outcome leads to an energy-efficient state that justifies the cost of the mechanism by reducing the number of large spines that need to be maintained. With their intricate structural complexity, large spines might require a more significant number of proteins, membrane traffic, and actin filaments to support their maintenance could be energetically costly. Thus, it indicates that activity-independent plasticity encourages homeostatic behavior by reducing the potential for higher energetic costs associated with maintaining large spines.

We have amended the sentence highlighted by the reviewer in the text to include this discussion in the revised discussion section. (lines 288 - 291)

- Methods pg 12, "the strength of that synapse cite". Presume there is a missing reference to be inserted instead of the word "cite"?

We are grateful to the reviewer for noticing this oversight, and we have added citations Chen et al., 2013 and Bartol Jr et al., 2015 to this sentence as well as the bibliography.

- Methods p13, says "The OU process was used by Loewenstein et al. (2011) to model activity-independent plasticity in a framework with multiplicative, instead of additive, noise, and takes the following form". However the noise term in eq(6) is additive, not multiplicative.

We thank the reviewer for suggesting to clarify how we reference Loewenstein et al. (2011). We have now modified this sentence to clarify this point, and it now reads as: "This approach was previously also used in Loewenstein et al. (2011) to model activity-

independent plasticity in a framework with multiplicative noise. Here, we will be applying it in an additive manner:" (lines 419-421)

- Methods p14, (also eq2 in main text) I find the notation used for $\mu_{\log,i}$ and $\sigma_{\log,i}$ a bit misleading since i is indexing time, I initially thought that this implied that these parameters are also time-varying, however it seems they are actually just functions of V (which of course changes with time).

We agree with the reviewer that this notation may be misleading. We have removed the subscript i from μ and σ and instead introduced them as functions of V , as suggested by the reviewer (equations 1, 4, 12, 13, 14, and throughout the text of the revised manuscript)

- Methods pg 14, I may have misunderstood but there appears to be an inconsistency between eqs 11+12 vs 13+14. The $\mu_s(V)$ in eq 11 has a delta term on the r.h.s., but in eq 13 both μ and δ are present in the numerator on the r.h.s. Is the delta term absorbed into μ_s or not?

We thank the reviewer for noting the inconsistency between the two sets of equations, which we have now fixed. The delta serves the purpose of shifting the data to be positive, which of course, the standard deviation already is (so no delta was required for this quantity). On the recommendation of another reviewer, equations (11) and (12) have now been moved to the results section and renumbered as equations (2) and (3). Additionally, the reviewer's comment brought to our attention that the location parameter in the lognormal equations needed to be a negative delta in order to subtract the positive shift we incurred by adding it to the sample mean. This has been corrected in equations 1, 4, and 14.

- Methods pg 15, the text in the "Details of the data set" section implies it will link to the source for the data and code but the link is not provided there at all, but in a later section "Data and code availability and history".

Thank you for spotting this. We have now removed the sentence that implied the link would be supplied here.

- Figure S1a caption says "black boxes", but these are not observable in the image.

We thank the reviewer for noting this mistake. We have removed the references to the black boxes. On the recommendation of another reviewer, we have reordered the supplemental figures to appear in the order in which they are referenced in the text. Therefore, Figure S1 has now become Figure S4.

- It might be worth also citing the following two papers who found that large spines were more stable than small spines via longitudinal in vivo 2-photon imaging:

Grutzendler, J., Kasthuri, N. and Gan, W.B., 2002. Long-term dendritic spine stability in the adult cortex. *Nature*, 420(6917), pp.812-816.

Holtmaat, A.J., Trachtenberg, J.T., Wilbrecht, L., Shepherd, G.M., Zhang, X., Knott, G.W. and Svoboda, K., 2005. Transient and persistent dendritic spines in the neocortex in vivo. *Neuron*, 45(2), pp.279-291.

- And potentially also of interest are the following theoretical papers which propose biophysical mechanisms by which larger spines/synapses could be more stable than small spines/synapses:

Shouval, H.Z., 2005. Clusters of interacting receptors can stabilize synaptic efficacies. *Proceedings of the National Academy of Sciences*, 102(40), pp.14440-14445.

O'Donnell, C., Nolan, M.F. and van Rossum, M.C., 2011. Dendritic spine dynamics regulate the long-term stability of synaptic plasticity. *Journal of Neuroscience*, 31(45), pp.16142-16156.

Bell, M.K., Holst, M.V., Lee, C.T. and Rangamani, P., 2022. Dendritic spine morphology regulates calcium-dependent synaptic weight change. *Journal of General Physiology*, 154(8), p.e202112980.

Jozsa, M., Donchev, T.I., Sepulchre, R. and O'Leary, T., 2022. Autoregulation of switching behavior by cellular compartment size. *Proceedings of the National Academy of Sciences*, 119(14), p.e2116054119.

We agree with the reviewer that the above citations would make valuable additions to the discussion on the stability of large spines. We have added these citations and descriptions of the key results to the revised manuscript in lines 313 - 316.

Typos

- page 2, parentheses oddly placed: "(e.g.) lateral diffusion, active trafficking, endocytosis, and exocytosis(Choquet and Triller, 2013; Ziv and Fisher-Lavie, 2014))"

- There is something ungrammatical about this sentence: "However, given the fact that synaptic changes are driven by molecular processes that are inherently noisy (e.g.) lateral diffusion, active trafficking, endocytosis, and exocytosis, implies that such spontaneous changes are inevitable."

- pg 4 "a positive mean change and a have smaller standard deviation", extra "a".

- pg 7 "are not significantly from", missing "different"

- pg 8 " during for the homo-", extra "for"
- pg 8 "that the spines became are more", extra "are"
- pg 10, "Our model relies a fast and slow mechanism", missing an "on"?
- pg 10, "fast oscillatory changes may preventing", -> prevent?
- pg 10, "and thus have no little to no", extra "no".
- pg 12, "in the case were these", were -> where
- pg 14, "where we use set the shift", extra "use"
- pg 12, "Briefly, he spine ROI", he -> the

We thank the reviewer for carefully reading our manuscript. We have corrected these issues in the revised version. We have also carefully reviewed the text to ensure no further typos are present.

We want to thank reviewer 2 for their interesting and in-depth questions in response to our original manuscript. Their valuable comments and contributions have provided new insights that have strengthened the paper and made it more accessible to all readers.

Reviewer #3 (Remarks to the Author):

In this study the authors imaged dendritic spines of GFP-expressing CA1 pyramidal neurons in hippocampal organotypic slices in the presence of TTX, picrotoxin and no added Mg²⁺. Images were obtained at 8 time points for overall durations of 55 minutes. In the course of these experiments, a subset of spines was stimulated (potentiated) by 2-photon glutamate uncaging or sham-uncaging (that is in the presence of absence of MNI-glutamate) after the second time point. Volumes of individual spines were then estimated at each of the time points. The authors then step through a series of increasingly elaborate stochastic models aimed to capture the dynamics of size changes, size distributions and the effects of stimulation on these dynamics and distributions, in order to obtain a new understanding of how both spontaneous synaptic fluctuations and local plasticity contribute to population-level synaptic dynamics.

There are many things to like about this study, in particular, the focus on short term dynamics. That is, while there have been quite a few attempts to model the activity independent size dynamics of dendritic spines over many hours and days, few studies have focused on the short-term dynamics (minutes). Moreover, the incorporation of rapid potentiation events into such models has not been attempted as far as I know.

Having said this, there are numerous matters that to my mind need to be addressed.

1) It would be much easier to follow the text if the explicit models used for Fig. 2 were placed in the Results rather than in the Methods section (Numerical), as essentially, the rationale for the development of the models and their evaluation appears twice – once in the main text and once in the Methods. Consequently, one has to jump back and forth between the two sections, which make the logic rather difficult to follow, in particular given the use of multiple terms for the same components (see below). I would suggest to merge everything into the Results, into a single coherent exposition.

We thank the reviewer for this valuable comment and suggestion. We agree that merging the two sections would make the text easier to follow. Within the constraints of the word limit of the main section, we have moved the most relevant part of the methods section, namely the description of the linear fits, to the results section, which was indeed discussed twice. This also led to a rearrangement of some sections covering the numerical methods, including an earlier mention of the transformations of sample statistics to lognormal statistics. We hope this improves the accessibility of our manuscript.

2) I found the model to be fairly complicated in comparison to prior models. In the models put forward in the past (e.g. the Kesten process (Statman et al., 2014), or a non-linear Langevin process (Yasumatsu et al., 2008; Hazan and Ziv, 2020), log-normality (or something close to it) emerged from a set of simple assumptions backed by data. Here, unless I am missing something, single time point changes are assumed a-priori to have log-normal distributions. Consequently, the model “plugs in” log normality through assuming that size changes are sampled from a log-normal distribution calculated separately (or approximately so) for each synapse, based on its momentary size. I am not sure I follow the rationale for this elaborate modeling process (which might relate to point 1 above).

We thank the reviewer for this helpful comment. We aimed to capture multiple experimentally observed features at once within a common modeling framework that have not been considered together but are all jointly present in our experimental data: lognormality, negative correlations, size dependence of changes, and global stability. These features are summarized in Figure 1. For each observed phenomenon, we have introduced a minimal mathematical description that captures them individually and then found a way to combine them. Even though prior models exist that describe these individual features, they may not represent the other features. For example, the classical models giving rise to lognormal distributions do not consider negative correlations on short-time scales. Similarly, experiments and models describing the properties of small vs. large spines do not consider lognormality, nor do they give rise to negative short-term correlations in spine changes. Therefore, we aimed to generate an effective model description step-by-step that was capable of reproducing the findings of four main experimentally observed features within a common modeling framework (Eq. 4). Finally, we also strived to include activity-dependent plasticity within this model to provide a common framework that made use of the same mechanisms for both types of plasticity.

3) Figure 1b and d. These fits are based on prior studies, among which is Hazan and Ziv 2020. In that study, a linear relation was (predicted and) observed between the *variance* and synaptic size *squared*. This is important if the function has a constant term, which the less-than-perfect fit to a linear function in Fig 1b suggests it does. One wonders if the fit would be better if the values were extracted from the squared function, allowing to do away with the need for “altering the “best” linear fits of the means and standard deviations”

We thank the reviewer for this interesting comment. We had indeed initially studied the relationships of the variance/mean with the spine size squared, as done in Hazan and Ziv. However, we found that the standard deviation/mean with the spine size led to similar results (including a need to introduce altered fits, which lowered the standard deviation and raised the mean). In our manuscript, we chose to use the standard deviation, mean, and spine size, which were all of the same order (μm^2). However, we agree that incorporating the analysis based on these squared quantities would be valuable. To this end, we have added an extra sentence in the introduction of the model that reads: “We note that previous work (including that of Hazan and Ziv, 2020), found linear relations between the spine size squared and the variance and mean. We saw that such fits were equally effective as the fits presented here, and leading to similar results (see Fig. S5).” (lines 128-130)

Figure S5 is a new supplemental figure which shows the same analysis and models as in Figure 2 but using the variance and spine size squared:

Figure S5: Linear fits can also be found between the variance (σ^2) and mean (μ) and the square of the spine sizes (V^2). a) Linear fits between the square of the spine sizes and sample change means, g_μ , and variances, g_{σ^2} , of activity-independent plasticity show good agreement. b) Similar to the earlier fits in Fig. 2, simulations using the linear fits from a) do not result in a stable distribution. The inset represents the simulated mean, which decreases significantly. c) The correlation obtained from one example step of the best fits log-normal simulations. The value of the slope is ≈ 0.1 , which is smaller than the correlations required. d) Altered linear fits for the mean and variance are used to achieve modeling goals. e) Distribution obtained from the simulation when the altered linear fits of the sample mean and standard deviation are used. The stability of the distribution is achieved as well as that of the mean (inset). f) The correlation obtained from one example step of the altered fits of log-normal simulations. The value of the slope is ≈ 0.1 , which is smaller than the correlations required.

4) This model adds a component which was not incorporated into prior models, the “negative momentum”. If I understand correctly, this term effectively implies that each synapse has some unique “set point”, or “short term memory” that acts to counter random changes on short time scales (minutes). Is this so? It is interesting that on longer time scales, no evidence was found for this component (Fig. 1F; see also Minerbi et al., 2009, Fig. S7). Could the authors add a plot similar to Fig. 1g for longer time intervals?

We are grateful to the reviewer for raising an interesting point. Yes, we agree that the negative momentum term endows the simulated synapse with a sort of “short-term memory”. In this way, it is induced to oppose previous changes, as we observe in the data. We hypothesize that it is a possible mechanism to avoid uncontrolled growth. We are aware that at longer timescales, this effect is not as strong or potentially absent

(Minerbi et al. 2009). Interestingly, Yasumatsu et al. 2008 reported a slight negative correlation (-0.189) between time points (with a time difference of 24 hours) and they have noted that a “slight correlation in control may represent a systematic trend of activity-dependent plasticity in our slice-culture conditions” where control refers to “without NMDA inhibitors” (see Figure 9C from Yasumatsu et al. 2008, below). When we studied the changes in spine size over 20 min time intervals with our dataset, we still found robust anti-correlation between consecutive snapshots (see figure below); however, this correlation slightly decreased for larger time bins, an observation which may help understand differences in experimental reports. We have added this figure to supplemental Figure S3, which shows the correlation between the timesteps that are spaced 17 to 20 min apart. We have also added a comment relating to the bin size-dependent negative correlation to the revised manuscript. Finally, in the description of potential future studies, we discussed this negative momentum term as a possible avenue for future research. (lines 607-630).

Figure 9. Relevance of the diffusion approximation for spine dynamics. **A, C**, Relationship of changes in W_i between the two successive time intervals in the presence (**A**) or absence (**C**) of NMDAR inhibitors. The correlation coefficients were -0.004 ($p = 0.969$) and -0.189 ($p = 0.054$), respectively. Values of W_i were obtained as described in Appendix A (S11). **B, D**, Probability-density distributions of changes in W_i per 1 d in the presence (**B**) or absence (**D**) of NMDAR inhibitors.

Figure S3i: In contrast to Fig. 1g, where consecutive size changes were compared, here we find all time point differences that are ≈ 20 minutes and compare these against each other. We see that points immediately following each other (highlighted by black squares) are negatively correlated even over this extended time period.

5) Equation (2) is the model ultimately derived in this manuscript. It would be useful to stick to the same terminology for each of its components as this would profoundly facilitate comprehension. For example, the first component is referred to as “Long-term stochasticity” but also as “linear fits”, “stochastic log-normal component”, the second is referred to as “drift” but also as “amendments to the fits” (and in Fig 4f “Short-time drift change”), and the third term is referred to as “Correlation” but also as “negative momentum” and “mean-reverting” (unless, admittedly, I failed to understand – please see point #1).

We thank the reviewer for the suggestion. We have now corrected all terms to be consistent throughout the manuscript as follows: "stochastic log-normal component" for the log-normal term, "Drift" for the global OU component, and "negative momentum" for the short-term correlation-promoting term throughout the revised manuscript.

6) In the attempt to alter the model parameters to fit the stimulation time point, an assumption is made that “the correlation term is not active during the stimulation”. What is the justification? This feels rather arbitrary. In addition, in the supplementary discussion it is stated that “As we did not need to alter the correlation component, we assume, intuitively, that this component is unaffected by the stimulation” so this is a bit confusing: Was it ultimately active or inactive?

We agree with the reviewer that the assumption of inactivity of the negative momentum term could have been presented more clearly in the original manuscript. We assume that the negative momentum term represents an inherently activity-independent process. By activating this term during the stimulation step, we would be implying that the previous stochastic activity-independent plasticity directly affects the subsequent activity-dependent change, either by increasing or decreasing it. For simplicity, we chose not to include this implication but instead deactivate the negative momentum term during the stimulation phase in the model. We have rephrased in the revised manuscript (lines 229-234) as follows:

"Additionally, we assume that the negative momentum term is a term that is inherent to activity-independent plasticity, i.e., it occurs as a stabilization mechanism and counters the previous stochastic change. As stimulation is a directed activity, negative momentum would hinder the growth of spines after stimulation by promoting shrinkage and imply that the previous stochastic activity-independent plasticity directly affects the subsequent activity-dependent change. Consequently, we choose to deactivate this term in the model during the stimulation step to avoid this scenario. However, future studies could consider including this or a generalized negative momentum term and study its role for the resulting synaptic size distribution."

7) "For the homosynaptic spines in Fig. 4d a different behavior emerges. We see that the standard deviation is tilted upwards, meaning that the resulting log-normal distribution has increased its standard deviation and that the spines become more variable during stimulation. This increase in the standard deviation is only observed in the medium-sized bins and not for the small or large spines". A) Given that synapses are rapidly enlarged by the potentiation protocol, isn't the increase in the standard deviation a trivial/expected outcome? B) Given that the Methods section states that "Medium-sized spines with a clear spine head within the field of view were preferentially targeted for stimulation", wasn't the selective effect on medium sized synapses expected?

We thank the reviewer for raising these two important points. In response to A), we agree that the increase in standard deviation due to stimulation is indeed expected and should have been mentioned in the text. We have corrected this by adding a comment (lines 219-221) in the revised manuscript as follows:

"We note that this increase follows intuitively for the following reasons: as the spines rapidly enlarged by the potentiation protocol, their variance will also be increased because i) they have grown beyond the normal size of activity-independent plasticity and ii) they are now large spines, which have been demonstrated to possess larger variance than small spines."

In response to B), the variance increase is observed in both the homosynaptic and heterosynaptic spines (see Figs. 4c and d). Additionally, Fig. 4d refers to the entire set of stimulated spines, which also includes initially small and large spines. When comparing the medium and small/large spines, we note that the standard deviation does not increase to the same extent, leading us to believe that the increase in standard deviation is not a selective effect. Instead, we believe that stimulated medium spines now exhibit the characteristics of those large spines, including an increased variance as they have grown to be the size of large spines.

We have added a corresponding comment (lines 222-224) to the manuscript.

8) Discussion – Indeed, prior studies assumed ergodicity, but the model proposed in the current study is not free of it. For example, the "fits to mean changes" (e.g. Fig 2b) imply that synaptic sizes converge toward some mean size, irrespective of their specific identity. Also, the changes they experience (the Long-term stochasticity component) are obtained from distributions that only depend on momentary synaptic sizes. Similarly, the drift term in equation 2 implies something similar. If I understand correctly, the departure from ergodicity is mainly to be found in the Correlation component. This is interesting and perhaps worthy of discussion. It is also worth mentioning that given the short observation periods, the lack of evidence for ergodic behavior is not completely unexpected.

We thank the reviewer for the comment; indeed, this is correct. By introducing an additional dimension, we can reformulate our model as a 2-dimensional (OU-like) Markov process:

$$V_{i+1} = V_i + \text{Logn} \left[\mu_{\log}(V_i), \sigma_{\log}(V_i), -\hat{\delta} \right] - \tilde{\theta}(V_i - \tilde{\mu}) - \theta(V_i - V_{i-1})$$

we define a new variable (vector) $\mathbf{x} = (V_{i-1}, V_i)$, we can rewrite the above equation as

$$\mathbf{x}_{i+1} = A \mathbf{x}_i + \xi(\mathbf{x}_i)$$

or, explicitly

$$\begin{pmatrix} V_i \\ V_{i+1} \end{pmatrix} = \begin{pmatrix} 0 & 1 \\ 1 - \bar{\theta} - \theta & \theta \end{pmatrix} \begin{pmatrix} V_{i-1} \\ V_i \end{pmatrix} + \begin{pmatrix} 0 \\ \bar{\mu}\bar{\theta} + \text{Logn} \left[\mu_{\log}(V_i), \sigma_{\log}(V_i), -\hat{\delta} \right] \end{pmatrix}$$

Given the nature of the space in which the model operates ($\mathbb{R}^+ \times \mathbb{R}^+$) and the absence of a dependence on time of the process coefficients, upon reaching stationarity, the process also becomes ergodic.

Under this model, all the terms, including those related to the negative-momentum, depend only on the current state of the process. This ergodicity assumption is also underlying existing models in the field.

The point that we considered worth highlighting in our paper, and that we hope is now formulated more precisely, is that to 55 minute window of the observation periods of our data, we cannot exclude a priori that a non-ergodic component within the experimentally reported spine-size dynamics is present since we can not test using our data (recorded over the limited time period) whether each spine explores all the points in the size space. The new sentence (lines 278-280) reads:

“In accordance with previous literature, we followed the ergodic hypothesis for our modeling. However, due to the 55 minute recording window in our data set, we could not test ergodicity directly or show that each spine explores the full phase space (see Fig. S3c,d).”

Minor comments –

1) The term “turnover” should probably be avoided as neither synapse not synaptic molecule turnover was studied here.

We agree with the point the reviewer makes and have removed one case of “turnover” and, in the other case, changed it to “baseline”.

2) Some justification for the use of the terms homosynaptic and heterosynaptic should probably be provided.

The reviewer is correct in their assessment that some clarity with respect to the homosynaptic and heterosynaptic spines should be provided. In the revised manuscript, we have added an additional description of these terms (lines 168-169):
"... those that have been stimulated (homosynaptic, i.e., those synaptic targets which have specifically been targeted for sLTP) and those that are left untouched (heterosynaptic, i.e., spines on the same dendritic stretch that are not directly potentiated)"

3) The methods mention that "For a subset of experiments, calcineurin or CaMKII was inhibited with FK506 (2 μ M, Tocris) or myristolated Autocamtide-2-related inhibitory peptide (AIP, 5 μ M, Calbiochem) respectively". Is this information correct? Did these agents affect the results?

We thank the reviewer for spotting this. We rechecked that these reagents were not applied to the cells from which our data sets were derived. This statement appeared here erroneously because the data set we used in this study (<https://github.com/meggl23/SpontaneousSpines>) contained not only the data we analyzed here but also included selected cells on which additional drugs were applied. Since these data sets were not considered, we have removed all references to calcineurin and CamKII blockers from the methods section.

4) Supplementary discussion – it is suggested that the correlation term acts as a fast time-scale mechanism which is only overcome with activity-dependent changes that allow the spines to grow into the sizes that fall on the long tail of the distribution. I note here that heavy tail distributions of spine sizes emerge in networks that never experienced any activity whatsoever (see for example Hazan and Ziv 2020), so some care should be made in this regard.

We thank the reviewer for this suggestion and agree that activity-dependent plasticity in itself does not generate a heavy tail. We have modified the statement in the following way "The stimulation induced forces allow for a collective shift of the spines to larger sizes." (lines 616-617)

REVIEWERS' COMMENTS:

Reviewer #1 (Remarks to the Author):

I am satisfied with the thorough revisions the authors have made to the manuscript, and believe it will be an excellent addition to the literature.

In reading it once more, I have a couple more minor, and optional, comments for the authors.

Page 15, red text before Eq. 12 and after Eq. 14, "log-normal" should be "log-normal distribution"

Page 11, red text beginning 6 lines from bottom with "Model justifications..." Grammar should be corrected to "...spines has been discussed by Shouval...and by Bell...and also by Jozsa..."

Page 4, model point 2. Writing would be clarified by changing to "...end up at the distribution of (Fig. 1c), which is stable over the timescales we consider..."

End of first paragraph of Introduction, I would suggest adding one more recent molecular mechanism reference to Bliss 1973 and Redondo 2010. One possibility is "How can memories last for days, years, or a lifetime? Proposed mechanisms for maintaining synaptic potentiation and memory. Smolen P, Baxter DA, Byrne JH.

Learn Mem. 2019 Apr 16;26(5):133-150."

Paul Smolen

Reviewer #2 (Remarks to the Author):

I found authors' responses to my comments very convincing and the manuscript is much improved. I noticed only one remaining minor typo:

- the new notation for μ_{\log} and σ_{\log} is clearer to me. However it looks like the authors may have inadvertently left one case of the old notation in near line 125: "...To determine the dependence of $\mu_{\log,i}$ and $\sigma_{\log,i}$ "

Reviewer #3 (Remarks to the Author):

The authors have addressed most of the points I raised satisfactory.

I suggest to consider these three minor points

1) Fig 1e legend - "The sum of all spine changes" - I believe the authors mean the collection, or the population, or the distribution, not the sum (as in an arithmetic sum).

2) The Methods state that "Medium-sized spines with a clear spine head within the field of view were preferentially targeted for stimulation". I understand the rationale provided in the rebuttal letter regarding the comment I made on this in the original review, but still, this is an important experimental detail that at least should be acknowledged in the results or discussion when mentioning the changes that were unique to the medium-sized spine population.

3) In the section "Links to biological mechanisms", the suggestion that "the presented log-normal model can be conceptually linked to the process known as geometric Brownian motion" points to the formulation of this process as equation 17. Unless I am missing something, the formulation of equation 17 appears to be identical to the model explored intensively by Yasumatsu and colleagues

(2008) (their equation 1), and later shown to be interchangeable under certain conditions with the Kesten process (Hazan and Ziv, 2020; equation 13). Consequently, perhaps the statement "Applying such a framework to the spine sizes will be a subject of future studies" should be reconsidered.

Dear Referees, dear Dr. Chong,

We once more thank the reviewers for the helpful feedback and the appreciation of our work. The comments have significantly improved the revised manuscript. In this response, we address the remaining comments below and provide a point-by-point reply (in bold).

Reviewer #1 (Remarks to the Author):

I am satisfied with the thorough revisions the authors have made to the manuscript, and believe it will be an excellent addition to the literature.

In reading it once more, I have a couple more minor, and optional, comments for the authors.

Page 15, red text before Eq. 12 and after Eq. 14, "log-normal" should be "log-normal distribution"

We thank the reviewer for this helpful point, which we have now corrected as suggested.

Page 11, red text beginning 6 lines from bottom with "Model justifications..." Grammar should be corrected to "...spines has been discussed by Shouval...and by Bell...and also by Jozsa..."

We are grateful to the reviewer for spotting this grammatical error, we now changed the sentence accordingly:

"... large spines has been discussed in Shouval (2005)⁴⁸ that proposed a mechanism based on clusters of interacting receptors in the synaptic membrane, Bell et al. (2022)⁴⁹ who considered a reaction-diffusion model of calcium dynamics and Josza et al (2022)⁵⁰ that showed ..." (lines 308-309)

Page 4, model point 2. Writing would be clarified by changing to "...end up at the distribution of (Fig. 1c), which is stable over the timescales we consider..."

We agree that this would be helpful to our readers and have modified the sentence as the reviewer proposed. (lines 102-103)

End of first paragraph of Introduction, I would suggest adding one more recent molecular mechanism reference to Bliss 1973 and Redondo 2010. One possibility is "How can memories last for days, years, or a lifetime? Proposed mechanisms for maintaining synaptic potentiation and memory. Smolen P, Baxter DA, Byrne JH. Learn Mem. 2019 Apr 16;26(5):133-150."

We agree with reviewer 1 that a more recent reference would be valuable and have included the suggested citation to Smolen et al. (2019). (line 26)

In summary, we thank reviewer 1 again for their valuable comments which helped us improve the accessibility and clarity.

Reviewer #2 (Remarks to the Author):

I found authors' responses to my comments very convincing and the manuscript is much improved. I noticed only one remaining minor typo:

- the new notation for μ_{\log} and σ_{\log} is clearer to me. However it looks like the authors may have inadvertently left one case of the old notation in near line 125: "...To determine the dependence of $\mu_{\log,i}$ and $\sigma_{\log,i}$ "

We are grateful to the reviewer for noticing this typo, which was indeed a remainder from the old notation we had overlooked. This has now been fixed by removing the subscript i. (after equation 1)

We want to thank the reviewer again for all the helpful comments, which helped us improve the manuscript.

Reviewer #3 (Remarks to the Author):

The authors have addressed most of the points I raised satisfactory.

I suggest to consider these three minor points

1) Fig 1e legend - "The sum of all spine changes" - I believe the authors mean the collection, or the population, or the distribution, not the sum (as in an arithmetic sum).

We thank the reviewer for catching this error. Indeed, we meant to refer to the collection of spine changes, not the arithmetic sum. We have now replaced sum with "collection" in the legend of Fig. 1e.

2) The Methods state that "Medium-sized spines with a clear spine head within the field of view were preferentially targeted for stimulation". I understand the rationale provided in the rebuttal letter regarding the comment I made on this in the original review, but still, this is an important experimental detail that at least should be acknowledged in the results or discussion when mentioning the changes that were unique to the medium-sized spine population.

We thank the reviewer for this comment, and have updated the corresponding sentence when we mention the changes of the medium-sized spine population:

"Additionally, in this current study, medium-sized spines were preferentially chosen for stimulation, as previous studies have shown that this population are most labile in terms of potentiation (for example Matsuzaki et al., 2004). It is possible that had we chosen to selectively target groups of large mushroom spines, or alternatively filopodia, that the outcome could be different." (lines 217-220)

3) In the section "Links to biological mechanisms", the suggestion that "the presented log-normal model can be conceptually linked to the process known as geometric Brownian motion" points to the formulation of this process as equation 17. Unless I am missing something, the formulation of equation 17 appears to be identical to the model explored intensively by Yasumatsu and colleagues (2008) (their equation 1), and later shown to be interchangeable under certain conditions with the Kesten process (Hazan and Ziv, 2020; equation 13). Consequently, perhaps the statement "Applying such a framework to the spine sizes will be a subject of future studies" should be reconsidered.

The reviewer is correct in their assessment, and we thank them for noticing this. We have removed the reference to the future studies and instead added the sentence:

“Previous work in this direction includes the study by Yasumatsu et al. (2008)¹² and Hazan and Ziv (2020)²⁴, the later has shown that the model equation 17 is interchangeable under certain conditions with the Kesten process.” (lines 644-646)

Finally, we would like to thank the reviewer 3 again for the careful reading of our text and insightful comments (both here and previously) that have strengthened the paper and made it more accessible to all readers.